# Engineering AvidCARs for combinatorial antigen recognition and reversible control of CAR function

Benjamin Salzer [1,2], Christina M. Schueller [1], Charlotte U. Zajc [1,2], Timo Peters[3], Michael A. Schoeber [1], Boris Kovacic[1], Michelle C. Buri [1], Elisabeth Lobner [4], Omer Dushek [5], Johannes B. Huppa [3], Christian Obinger [6], Eva M. Putz [1], Wolfgang Holter[1,7], Michael W. Traxlmayr [2,6 ✉] & Manfred Lehner [1,2,7 ✉]

T cells engineered to express chimeric antigen receptors (CAR-T cells) have shown impressive clinical efficacy in the treatment of B cell malignancies. However, the development of CAR-T cell therapies for solid tumors is hampered by the lack of truly tumor-specific antigens and poor control over T cell activity. Here we present an avidity-controlled CAR (AvidCAR) platform with inducible and logic control functions. The key is the combination of (i) an improved CAR design which enables controlled CAR dimerization and (ii) a significant reduction of antigen-binding affinities to introduce dependence on bivalent interaction, i.e. avidity. The potential and versatility of the AvidCAR platform is exemplified by designing ON-switch CARs, which can be regulated with a clinically applied drug, and AND-gate CARs specifically recognizing combinations of two antigens. Thus, we expect that AvidCARs will be a highly valuable platform for the development of controllable CAR therapies with improved tumor specificity.

[1] St. Anna Children's Cancer Research Institute (CCRI), 1090 Vienna, Austria. [2] Christian Doppler Laboratory for Next Generation CAR T Cells, 1090 Vienna, Austria. [3] Center for Pathophysiology, Infectiology and Immunology, Institute for Hygiene and Applied Immunology, Medical University of Vienna, 1090 Vienna, Austria. [4] Department of Biotechnology, University of Natural Resources and Life Sciences, 1190 Vienna, Austria. [5] Sir William Dunn School of Pathology, University of Oxford, Oxford OX1 3RE, UK. [6] Department of Chemistry, Institute of Biochemistry, University of Natural Resources and Life Sciences, 1190 Vienna, Austria. [7] Department of Pediatrics, St. Anna Kinderspital, Medical University of Vienna, 1090 Vienna, Austria. ✉email: michael.traxlmayr@boku.ac.at; manfred.lehner@ccri.at

CARs are synthetic proteins consisting of antigen-binding domains linked to T-cell-activating signaling domains (Fig. 1a)[1]. T cells genetically engineered to express CARs (CAR-T cells) efficiently eliminate antigen-expressing target cells, which is reflected by the impressive efficacy of CD19-directed CAR-T cell therapies in patients with B cell malignancies[2]. A major challenge in translating this success to other malignancies is the fact that CAR antigens are almost exclusively tumor-associated antigens (TAAs), which are not truly tumor specific. Since most TAAs are also expressed in vital healthy tissues, efficient TAA targeting is expected to result in unacceptable on-target/off-tumor toxicity, which has indeed been observed in several clinical studies[3-7]. Ultimately, this could preclude increasing the efficacy of CAR-T cell therapies to levels required for sustained remission of solid tumors. The problem of on-target/off-tumor toxicity is further aggravated by the fact that the activity of CAR-T cells is difficult to control in a reversible manner.

To address the lack of control and tumor specificity associated with CAR-T cells, several promising strategies have been developed including CARs for combinatorial antigen recognition[8-14], affinity-tuned CARs to minimize killing of healthy cells expressing low levels of target antigen[15-20] and CARs whose function can be regulated by administration of small molecules or soluble proteins[10,21-29]. Despite these important advances, there is still a need for improvement. For example, currently available systems for the control of CAR-T cells are either based on small molecules which are suboptimal for clinical application or on suicide

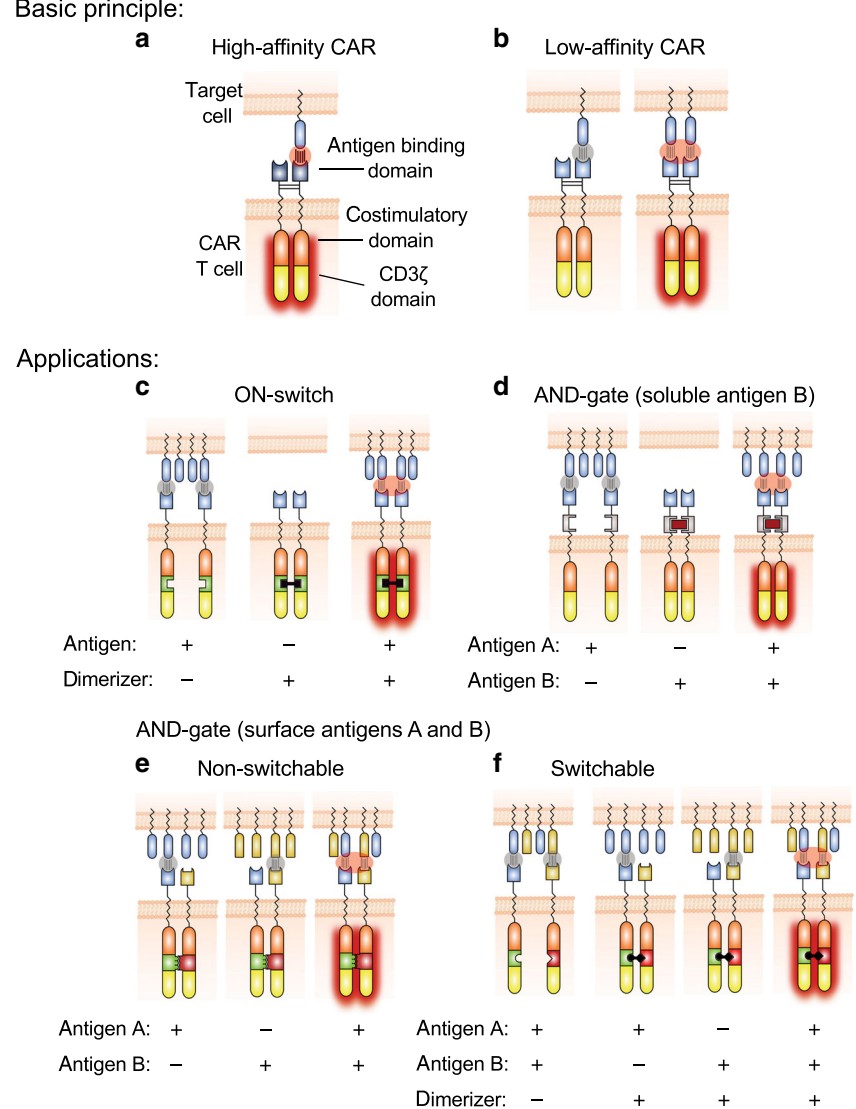

**Fig. 1 Principle and applications of AvidCARs. a** Schematic of a conventional CAR (high-affinity CAR) typically containing a high-affinity antigen-binding domain (usually a single-chain variable fragment, scFv) fused via a dimerizing hinge region (derived from, e.g., CD8α) to the cytoplasmic domains of a co-stimulatory receptor (mostly 4-1BB or CD28) and CD3ζ. Due to high-affinity binding, monovalent interaction with the target antigen is sufficient for CAR activation. **b** Schematic of a low-affinity CAR in which a low-affinity antigen-binding domain is fused to the same conventional CAR backbone shown in (**a**). We hypothesized that such low-affinity CARs require bivalent interaction with the antigen for efficient activation. We further hypothesized that this avidity-based concept (i.e., the dependency on bivalent antigen recognition) can be exploited for various applications: **c** for generating ON-switch AvidCARs whose function is regulated by a dimerizing small molecule; **d** for constructing AND-gate AvidCARs which are dimerized by a soluble antigen B, leading to avidity-based recognition of antigen A; **e**, **f** and for the design of heterodimeric AND-gate AvidCARs specifically recognizing a combination of two surface bound antigens with two different recognition domains, respectively. This AND-gate concept can be realized in a nonswitchable format by constitutive CAR heterodimerization (**e**) or in a switchable version by inducing heterodimerization with a small molecule (**f**).

switches which irreversibly destroy the CAR-T cells. Furthermore, previously described AND-gate CARs suffer from low specificity[12–14] or their inability to differentiate between a double-positive cell and two single-positive cells, which express the antigens A and B complementarily and are located in close proximity to each other[21,30]. That is, in this SynNotch strategy, expression of the CAR can be induced by healthy cells expressing antigen A, enabling recognition of nearby healthy cells expressing antigen B. Furthermore, the SynNotch system is based on xenogeneic and thus potentially immunogenic proteins.

One critical parameter defining the potency and function of CARs is the affinity of their antigen-binding domains. Importantly, the interaction with a target antigen not only depends on the affinity of an individual antigen-binding domain, but also on the valency of the interaction. That is, a multivalent interaction amplifies the individual affinities through a phenomenon known as avidity. Many types of natural immune reactions are based on low affinities, which are amplified by avidity effects, such as antigen binding by IgM during early humoral immune responses. However, the influence of avidity effects on CAR function has mostly been neglected in the CAR field so far, even though most currently used CARs are based on naturally dimeric components[31–33].

Here, we demonstrate that this aspect can be exploited for the integration of inducible and logic control functions into CAR molecules. We show that it is possible to generate avidity-controlled CARs (AvidCARs) which are highly potent and dependent on bivalent antigen engagement (Fig. 1). These AvidCARs are based on two main design principles: controlled CAR dimerization and low-affinity antigen binding. We exemplify the potential of the AvidCAR platform (1) by generating a controllable ON-switch CAR that can be regulated by CAR homodimerization with a clinically applied small molecule (Fig. 1c); (2) by constructing an AND-gate CAR that is triggered only in the presence of both a surface molecule on tumor cells and a soluble protein found in tumor stroma (Fig. 1d); and (3) by generating a heterodimeric AND-gate CAR which enables the combinatorial recognition of two different antigens exclusively when co-presented on the surface of the same cell. This latter AND-gate AvidCAR is designed either in a constitutive (Fig. 1e) or in a switchable version regulated by a dimerizing small molecule (Fig. 1f). Finally, we also investigate the suitability of different antigen-binding modules and various co-stimulatory domains for the design of efficient AvidCARs. Together, our study illustrates the potential of exploiting avidity effects and demonstrates that the AvidCAR platform is a versatile concept for inducible and combinatorial CAR control.

## Results

**Construction of avidity-controlled CARs.** Since a critical design principle of AvidCARs is the control over CAR dimerization, the usage of CAR components potentially causing uncontrolled dimerization or oligomerization needs to be avoided. Given the recent report on scFv-mediated CAR tonic signaling (probably caused by scFv-mediated clustering)[34], we looked for an antigen-binding domain based on a single protein domain, which prevents intermolecular dimerization as was observed with several scFvs[35–37]. Therefore, we chose the human epidermal growth factor receptor (hEGFR)-specific binder E11.4.1, which was previously engineered based on the single-domain protein rcSso7d[38]. As described above, we hypothesized that the construction of an AvidCAR also requires the incorporation of low-affinity binding domains (Fig. 1b). For this purpose, we reduced the affinity of the high-affinity binder E11.4.1 by performing an alanine scan, resulting in eight different E11.4.1 mutants. To determine the

affinities of those mutants, they were expressed as monomeric proteins (Supplementary Fig. 1a) and titrated on Jurkat cells expressing truncated hEGFR (hEGFRt, Supplementary Fig. 1c). These titration experiments yielded an affinity of 13 nM for the high-affinity binder E11.4.1, which is in close agreement with the previously published value (17 nM)[38]. Among the mutants we identified two (E11.4.1-G25A and E11.4.1-G32A) with substantially decreased, yet measurable affinities of 125 and 877 nM, respectively (Fig. 2a). Complementary affinity determination by surface plasmon resonance (SPR) analysis with the entire extracellular domain of hEGFR yielded highly comparable $K_d$ values (Supplementary Fig. 1b).

To test our hypothesis that only low-affinity, but not high-affinity CARs are dependent on bivalent interaction, we incorporated the high-affinity (E11.4.1; $K_d$ of 13 nM), intermediate-affinity (E11.4.1-G25A; $K_d$ of 125 nM), or low-affinity (E11.4.1-G32A; $K_d$ of 877 nM) hEGFR binders into CARs comprising the hinge and transmembrane region of CD8α, followed by the signaling domains of 4-1BB and CD3ζ (termed BBz-CAR, Fig. 2b). In addition, each of those three CARs was also created in a second version in which the two extracellular cysteine residues in the CD8α hinge region, which are known to cause CD8α dimerization[32], were mutated to serine residues, resulting in a CAR backbone termed Ser-BBz. All CAR variants were expressed at similar levels in primary human T cells (Supplementary Fig. 2a). In co-culture with hEGFRt-expressing target cells (Supplementary Fig. 2b) both the high- and intermediate-affinity CARs triggered target cell lysis and IFN-γ secretion both in the BBz-, as well as in the Ser-BBz-format, albeit at slightly higher efficiency in the dimeric BBz version (Fig. 2c and Supplementary Fig. 2c). In contrast, the low-affinity CAR only mediated lysis and IFN-γ production when expressed with dimerization-promoting cysteine residues in its CD8α hinge (Fig. 2c and Supplementary Fig. 2c), thus supporting our initial hypothesis that low-affinity CARs are dependent on bivalent interactions.

Next, we used a mathematical model to investigate whether the observed avidity-driven activation of AvidCARs is mediated by an increase in the effective lifetime of the CAR–antigen interaction (by allowing for increased rebinding) and/or by increased binding (i.e., more CAR occupancy by antigen). A longer lifetime is thought crucial for pMHC binding to the T-cell receptor (TCR) as a result of kinetic proofreading and this has recently been shown to be the case also for CARs[39]. Thus, to investigate how avidity alters CAR occupancy and lifetime, we set up a mathematical model of antigen binding to a bivalent CAR with kinetic proofreading (e.g., representing CAR phosphorylation) where a tunable parameter ($\sigma$) was used to modulate avidity (Supplementary Fig. 3a). The model shows that increasing avidity has only a modest effect on CAR occupancy but a dramatic effect on the amount of phosphorylated CAR, highlighting that increased avidity strongly prolongs the duration of individual CAR–antigen interactions, thereby enhancing the signaling capacity. In summary, this mathematical model predicts that increased avidity leads to improved antigen sensitivity (Supplementary Fig. 3b).

To investigate this dependence of AvidCAR sensitivity on avidity experimentally, we switched to another assay system based on small molecule-mediated CAR dimerization. For that purpose, the FK506-binding protein (FKBP)12 and a mutant FKBP12-rapamycin binding domain (FRB) were integrated into the Ser-BBz-CARs for small molecule (AP21967)-induced CAR heterodimerization (Fig. 2d). To additionally investigate the impact of antigen dimerization on avidity, we integrated the FKBP12-F36V domain into hEGFRt for small molecule (AP20187)-mediated antigen homodimerization (Fig. 2d). Primary human T cells expressing either high-affinity CARs or the corresponding low-

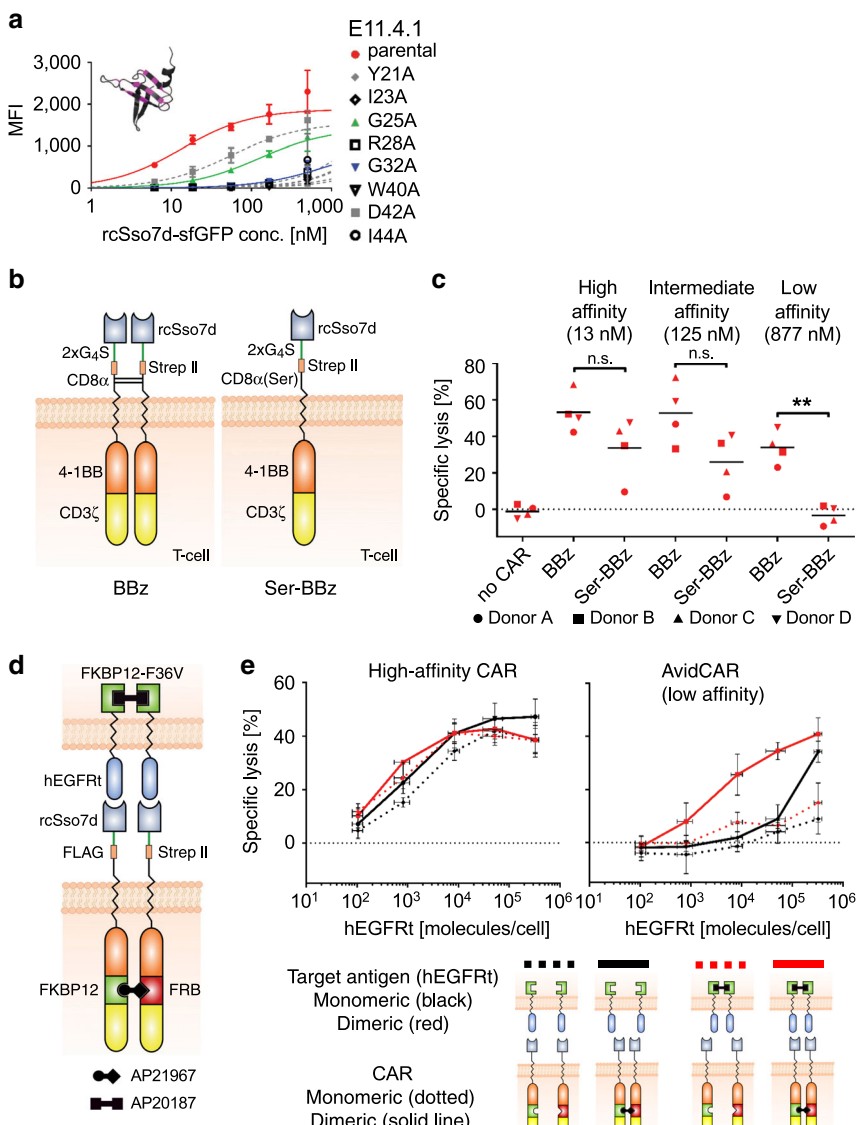

**Fig. 2 Generation of AvidCARs. a** Flow cytometric determination of the affinities of hEGFR-specific binder mutants. Soluble binder proteins (sfGFP fused to E11.4.1 or mutants thereof) were titrated on hEGFRt[pos] Jurkat cells. Shown are the median fluorescence intensities (MFI) of bound E11.4.1-variants (three independent experiments; mean values ± SD). Colored lines indicate the high-, intermediate-, and low-affinity binders used in the following experiments. The insert in the diagram shows the structure of the original Sso7d binder scaffold with its mutated binding surface in pink (PDB 1SSO). **b**, **c** Dependence of CAR function on binder affinity and CAR dimerization. **b** Schematic of the tested CARs. Selected rcSso7d-based binders were fused to a CAR backbone in which the CD8α hinge contained either two cysteine- or two serine residues instead (see SEQ-IDs 1 and 2 in Supplementary Data 1, respectively). **c** The figure shows the cytolytic activity of mock-T cells (no CAR) and T cells expressing the monovalent (Ser-BBz) and bivalent (BBz) CARs with different affinities against hEGFRt[pos] Jurkat target cells. (Luciferase-based assay; four independent experiments with four different donors; **p < 0.01, calculated by two-tailed paired t-test). **d**, **e** Influence of CAR and antigen dimerization on the activity of AvidCARs and their high-affinity versions. **d** Schematic of the experimental system for controlled dimerization of CARs (containing domains for conditional heterodimerization by the compound AP21967; see SEQ-IDs 3 and 4 in Supplementary Data 1) and for separately regulated dimerization of the antigen hEGFRt (containing a domain for conditional homodimerization by AP20187; SEQ-ID 21). **e** T cells, which expressed CARs with either high- or low-affinity binders (i.e., $K_d$ of 13 nM or 877 nM, respectively), were co-cultured with Jurkat cells expressing different levels of hEGFRt. The effect of CAR and antigen dimerization was tested by separately regulating these two dimerization events with AP21967 and AP20187, respectively. The figure shows the lysis of hEGFRt-expressing Jurkat cells as determined by a flow cytometry-based cytotoxicity assay (two independent experiments with three different donors; mean values ± SD). Exact p values are given in Supplementary Data 2. Source data are provided as a Source data file.

affinity AvidCARs were then co-cultured ± AP21967 (for CAR dimerization) and ± AP20187 (for antigen dimerization) with Jurkat target cells expressing the antigen hEGFRt at different densities ranging from hundreds to several hundred thousand molecules per cell (Supplementary Fig. 4a). Target cell lysis mediated by the high-affinity CAR was largely independent of the dimerization state of both the CAR and the antigen, strongly

suggesting that the high-affinity CAR was not dependent on bivalent interaction (Fig. 2e, left diagram). In contrast, the low-affinity AvidCAR efficiently triggered activation only in a dimeric form (Fig. 2e, right diagram, black and red solid lines), again demonstrating its dependency on bivalent interaction, i.e., avidity. Furthermore, the antigen sensitivity of this dimeric AvidCAR was strongly increased if also the antigen was dimerized (red solid line vs.

black solid line, Fig. 2e, right diagram). Thus, enhancing the avidity effect of the dimeric AvidCAR—mediated through additional antigen dimerization in this experiment—resulted in increased antigen sensitivity, as predicted by the mathematical model described above. Similar activation patterns were observed when measuring IFN-γ release (Supplementary Fig. 4b). High-affinity CAR and AvidCAR constructs were expressed at similar levels, hence excluding expression-based effects (Supplementary Fig. 4c).

Together, these experiments demonstrate that it is possible to engineer AvidCARs which depend on bivalent interaction and whose function can thus be regulated by drug-mediated dimerization. Moreover, the comparable efficacy of the monomeric and dimeric high-affinity CAR (Fig. 2e) is in line with a previous report showing that a single CD3ζ domain is sufficient for efficient signaling[40].

**scFvs may preclude controlled CAR dimerization**. Global efforts in the development and clinical testing of novel CAR-T cell constructs almost exclusively rely on scFv-based binders. To test the suitability of scFvs for generating AvidCARs, we incorporated the hHER2-specific low-affinity scFv 4D5-5 ($K_d$ of 1.1 μM)[15] into both the BBz- and the Ser-BBz-CAR (Fig. 3a). In contrast to the rcSso7d-based CARs described above, in the scFv-based CAR the Cys→Ser-mutation had little effect on target cell lysis and IFN-γ secretion by CAR-T cells co-cultured with hHER2t$^{pos}$ target cells (Fig. 3b), despite even lower expression levels of the CAR containing the Ser-mutation (Supplementary Fig. 5a). Given those functional differences compared to the rcSso7d-based CARs despite similarly low affinities, we hypothesized that scFv-mediated oligomerization might prevent the formation of strictly monomeric CARs. Dimerization of scFvs has previously been reported for soluble scFvs, resulting in the formation of so-called diabodies, in which the VH of one scFv molecule pairs with the VL of another scFv molecule[35–37] (Fig. 3c).

To investigate whether intermolecular dimerization of VH and VL between scFvs can also trigger CAR oligomerization, we expressed this scFv as an Fv, i.e., without the linker connecting VL and VH, thus preventing such intermolecular dimerization (Fig. 3d and Supplementary Fig. 5b). In this Fv-based CAR, the VH was fused to the Ser-BBz-backbone, whereas the VL was anchored via a separate CD8α(Ser) transmembrane domain without an intracellular signaling region. Importantly, compared to the scFv-based CAR, significantly reduced cytotoxicity was observed with the Fv-based CAR (Fig. 3e), strongly suggesting that removing the linker from the scFv prevents CAR dimerization and thereby avidity-based activation of low-affinity CARs. Of note, CAR activity could be restored by AP20187-induced homodimerization of two Fv-based CAR molecules (Fig. 3e). This demonstrates (1) that separate VH- and VL-containing constructs assemble into a CAR molecule with a functional Fv unit and (2) that in the absence of scFv-mediated oligomerization, avidity can be controlled by inducing dimerization. As expected, in the absence of the VL construct the dimerization of the VH-construct had no effect, confirming that antigen binding requires an assembled Fv comprising VH and VL (Fig. 3e). Furthermore, all constructs were expressed at similar levels, thus excluding expression-mediated effects (Supplementary Fig. 5c).

Together, these data demonstrate that intermolecular VH/VL dimerization of at least certain scFvs can cause CAR oligomerization. As a consequence, such clustering scFvs are undesired in our context not only because of tonic CAR signaling[34,41], but also because of preventing the avidity-based control of AvidCARs.

**Influence of different co-stimulatory domains**. Next, we investigated whether the dependence of low-affinity CARs on

dimerization is also observed with other co-stimulatory domains by exchanging the 4-1BB domain in the Ser-BBz-CAR backbone for the endodomain of CD28, ICOS, OX40, or CD2 in our hEGFR-specific rcSso7d-based AvidCAR as described above. All CARs additionally contained an FKBP12-F36V domain for conditional homodimerization by AP20187 (Fig. 3f). In a cytotoxicity experiment with primary human T cells expressing those CARs, significant dimerization dependence was observed with 4-1BB-, ICOS- and CD2-mediated co-stimulation, but not if CD28 or OX40 were incorporated (Fig. 3g). IFN-γ secretion was significantly induced by dimerization of 4-1BB- and CD2-based CARs (Supplementary Fig. 5e). Albeit not being significant, a trend was also observed with the endodomain of OX40. In summary, these data suggest that among the co-stimulatory domains tested, the ones derived from 4-1BB or CD2 are best suited for construction of efficient AvidCARs—at least with the binding affinities and antigen densities tested in these experiments.

**AvidCARs with AND-gate function**. Current bispecific CARs are characterized by high-affinity binding domains and, hence, do not possess AND-gate function but OR-gate function, i.e., activation in the presence of either antigen A or B or both[42–44]. Moreover, the homodimeric/-oligomeric nature of current CAR designs facilitates multivalent engagement of a single antigen by several identical binding domains, thus additionally precluding efficient AND-gate function. Here, we tested whether our AvidCAR platform, in which undesired homodimerization can be prevented, allows for the generation of bispecific CARs with robust AND-gate function. To this end, we generated an additional low-affinity binder recognizing a different model antigen. As a starting point, we chose the hHER2-specific single-domain binder zHER2-AK, which is based on the previously engineered affibody ZHER2:4[45]. High-affinity binding of zHER2-AK to hHER2 was confirmed by SPR, yielding a $K_d$ of 17 nM (Supplementary Fig. 6b). Again, to reduce its affinity, the affibody was subjected to an alanine scan. The resulting 12 mutants were incorporated into the Ser-BBz-CAR backbone comprising the FKBP12-F36V domain for functional screening by controlled homodimerization. In a cytotoxicity assay, the CAR containing the binder mutant zHER2-AK-R10A triggered lysis of hHER2t$^{pos}$ cells in a dimerization-dependent manner (Supplementary Fig. 6a), indicating that the reduced affinity of this mutant necessitates bivalent antigen engagement. Indeed, subsequent SPR analysis confirmed that this affibody-mutant binds to hHER2 with low affinity ($K_d$ of 865 nM), which is comparable to that of the hEGFR-specific rcSso7d-based binder E11.4.1-G32A described above.

To generate the hEGFR/hHER2-bispecific AND-gate AvidCAR, those two low-affinity binders were incorporated into two separate Ser-BBz-CAR constructs with complementary heterodimerization domains (Fig. 4a). As we had hypothesized, primary human T cells expressing this AND-gate AvidCAR efficiently triggered lysis of target cells expressing both hEGFRt and hHER2t, while lysis of cells expressing only hEGFRt or hHER2t was comparable to background lysis by CAR$^{neg}$ T cells (Fig. 4b and Supplementary Fig. 6c). This experiment clearly demonstrates the dependency of this heterodimeric AvidCAR on combinatorial antigen recognition. Moreover, cytotoxicity was only observed in the presence of the heterodimerizing agent, demonstrating that this AND-gate AvidCAR can be additionally controlled by administration of a small molecule. Of note, the incorporation of the high-affinity versions of both binders ($K_d$ values of 13 and 17 nM for E11.4.1 and zHER2-AK, respectively) abrogated the AND-gate character and the dependence on

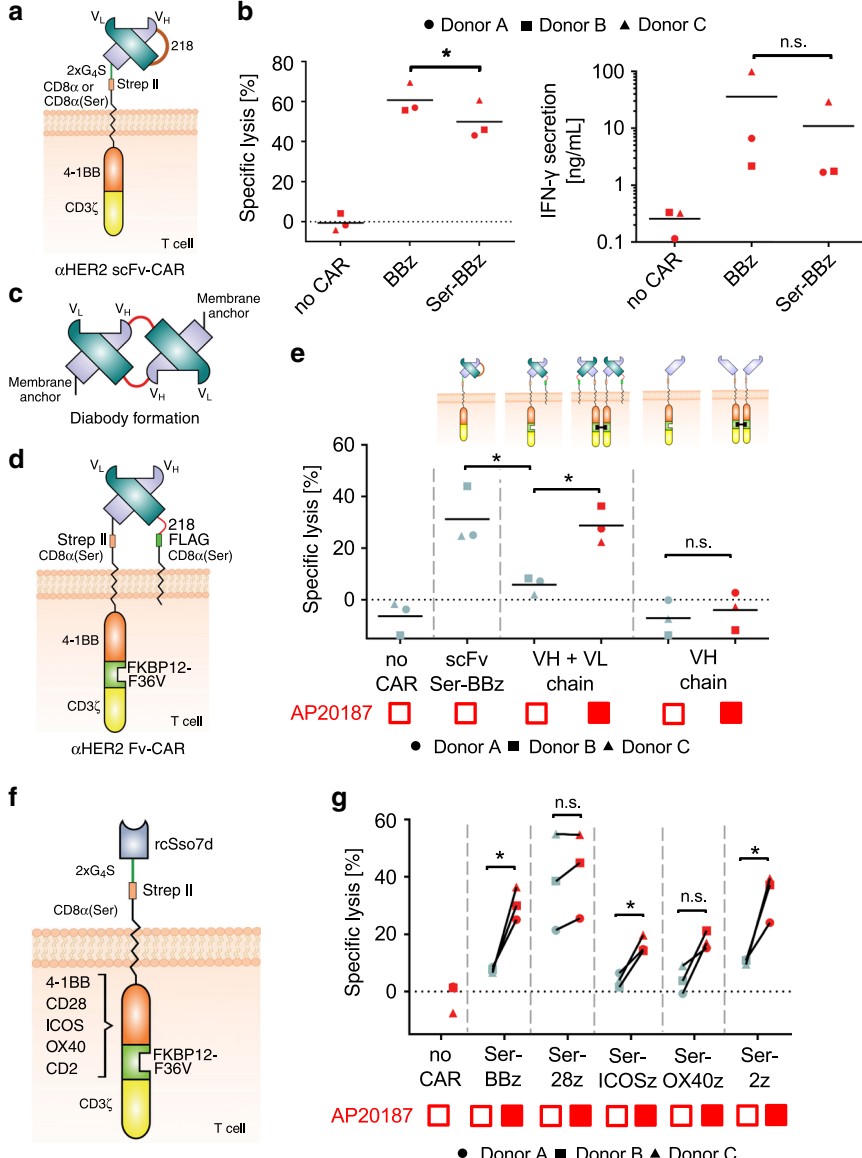

**Fig. 3 Influence of extra- and intracellular CAR domains. a, b** Clustering of a hHER2-specific scFv. scFv-containing CARs were either based on a BBz and Ser-BBz-backbone, as shown in (**a**) (SEQ-IDs 5 and 6, respectively). **b** The figure shows the cytolytic and secretory activity of mock-T cells (no CAR) and T cells expressing the BBz- or Ser-BBz-CAR in co-cultures with hHER2t^pos Jurkat cells [flow cytometry-based assay; three independent experiments with different donors; *$p < 0.05$, calculated by two-tailed paired $t$-test (left panel) or by two-tailed ratio-paired $t$-test (right panel)]. **c** Schematic of diabody formation of two scFvs by intermolecular heterodimerization of VHs and VLs. **d, e** Prevention of diabody formation enables defined CAR dimerization. **d** Schematic of a CAR with VH and VL on separate polypeptide chains for preventing diabody formation (SEQ-IDs 7 and 8). Both constructs are shown in the heterodimerized state together forming a hHER2-specific Fv-based CAR. **e** Cytolytic activity of mock-T cells (no CAR) or T cells expressing either the hHER2-specific scFv-CAR [shown in (**a**)] or the VH and VL constructs [shown in (**d**)] in co-cultures with hHER2t^pos Jurkat cells. AP20187 was added for homodimerization of the VH-construct as indicated (filled red boxes). [flow cytometry-based assay; two independent experiments with three different donors; *$p < 0.05$, calculated by two-tailed paired $t$-test and by two-tailed unpaired $t$-test (scFv Ser-BBz vs. VH + VL chain)]. **f, g** Influence of co-stimulatory domains. **f** Schematic of hEGFR-specific rcSso7d-based low-affinity CARs which contain different co-stimulatory domains. The 4-1BB domain of the AvidCAR was substituted by the endodomain of CD28, ICOS, OX40, or CD2 (SEQ-IDs 9, 10, 11, 12, and 13, respectively). The FKBP12-F36V domain was integrated for conditional homodimerization. **g** rcSso7d-based low-affinity CARs with different co-stimulatory domains were expressed in T cells and co-cultured with hEGFRt^pos Jurkat cells. The figure shows the cytolytic activity of the T cells and its dependence on AP20187-mediated CAR dimerization (flow cytometry-based assay; two independent experiments with three different donors; *$p < 0.05$, calculated by two-tailed paired $t$-test). Exact $p$ values are given in Supplementary Data 2. Source data are provided as a Source data file.

dimerization (Fig. 4b), thus confirming the necessity of low-affinity antigen binding. Both the high-affinity and the low-affinity heterodimeric CARs were expressed at similar levels, excluding any major expression-based bias (Supplementary Fig. 6d). Fusion of the two low-affinity binders in tandem into a single, monomeric CAR molecule (TanCAR) (Fig. 4a), also

yielded specificity for double-positive target cells (Fig. 4b), albeit without the possibility to conditionally control CAR function with a small molecule.

Together, these data demonstrate that bispecific AvidCARs can be engineered for robust combinatorial antigen recognition, which ultimately allows for specific elimination of target cells co-

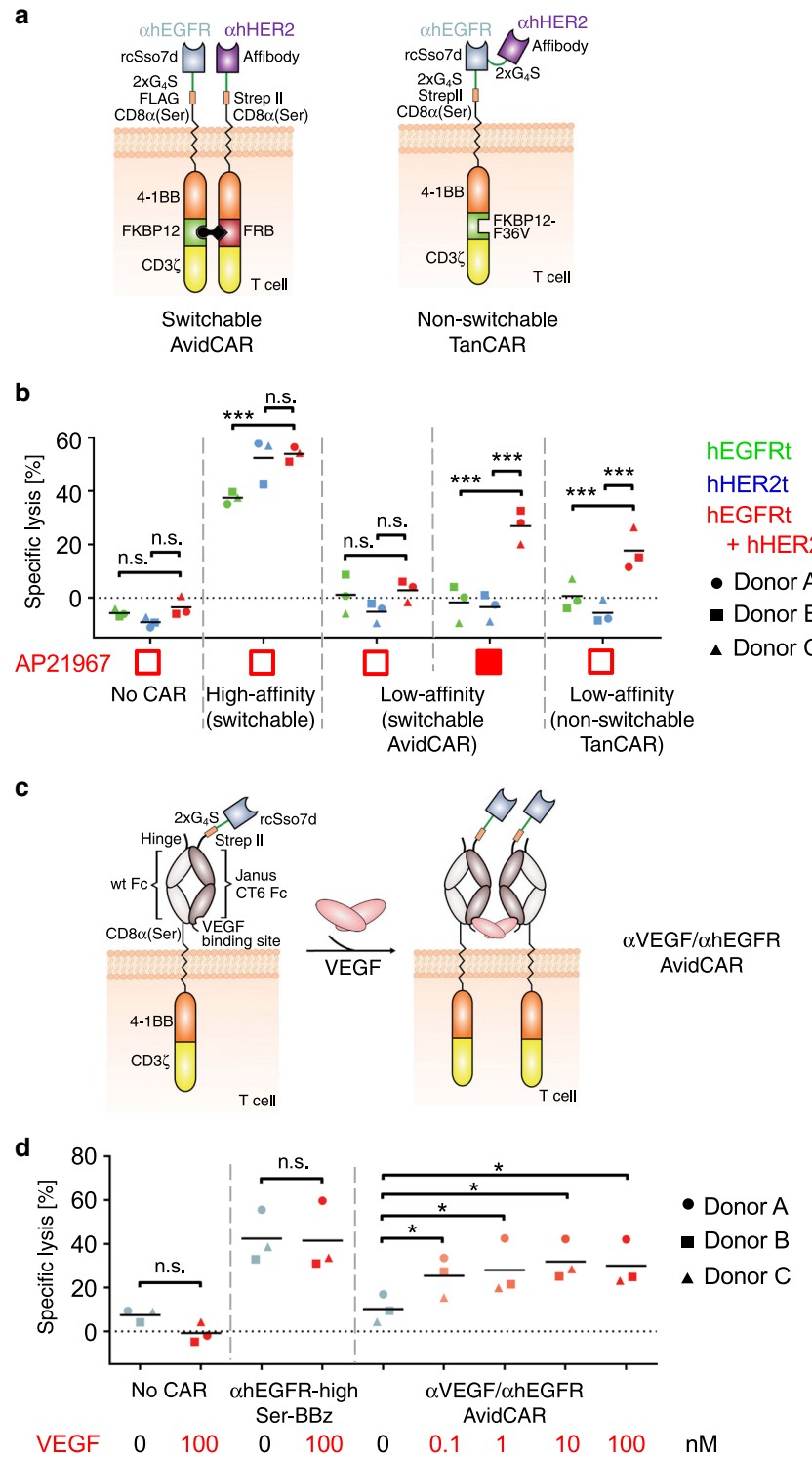

expressing two antigens on their surface, but not their single-positive counterparts.

**AND-gate with a surface and a soluble antigen.** Based on the data described above, we hypothesized that the AvidCAR concept also enables the construction of CARs that are specifically activated by combinatorial recognition of a surface antigen and a tumor-resident soluble factor, which in this case acts as a CAR-dimerizing agent. For that purpose, we designed a hEGFR-specific AvidCAR additionally containing the heterodimeric Fc antigen

binding (Fcab) mutant JanusCT6, which recognizes vascular endothelial growth factor (VEGF)[46] and expressed this AvidCAR construct in primary human T cells (Supplementary Fig. 6e). As VEGF is naturally present as a dimer, we hypothesized that it could homodimerize the AvidCAR molecules and thereby facilitate bivalent recognition of hEGFR (Fig. 4c). Indeed, VEGF triggered lysis of hEGFRt-expressing Jurkat cells, thus confirming our hypothesis (Fig. 4d). Of note, VEGF had no effect on target cell viability in the presence of T cells expressing either no CAR or a VEGF-independent high-affinity CAR, thus excluding any

**Fig. 4 Exploitation of the avidity effect for generating AvidCARs with AND-gate function. a, b** Construction of AND-gate AvidCARs for combinatorial targeting of two cell surface antigens. **a** Schematic of an hEGFR/hHER2-bispecific CAR in a switchable format (SEQ-IDs 14 and 15) and a nonswitchable TanCAR format (SEQ-ID 16). **b** Lysis of single- and double-antigen positive Jurkat cells by T cells expressing different hEGFR/hHER2-specific CARs (flow cytometry-based assay; two independent experiments with three different donors; ***$p < 0.001$, calculated by a mixed model, with CAR group, antigen, and their interaction as fixed effects and the donor as a random effect to take into account the correlated structure, unadjusted two-sided $p$ values are given). T cells expressing the hEGFR/hHER2-bispecific CARs shown in (**a**) were co-cultured with Jurkat target cells expressing either hEGFRt or hHER2t or both antigens. Mock-T cells (no CAR) or T cells expressing the high-affinity version of the hEGFR/hHER2-bispecific switchable AvidCAR served as negative and positive controls, respectively. AP21967 was added for heterodimerization of the switchable bispecific AvidCAR as indicated (filled red box). **c, d** Generation of AND-gate AvidCARs recognizing a membrane-bound antigen A and a soluble antigen B. **c** A heterodimeric Fcab mutant JanusCT6, which recognizes VEGF, was integrated into the hEGFR-specific low-affinity Ser-BBz-CAR (SEQ-IDs 17 and 18). Binding of VEGF, which is a homodimer, results in homodimerization of the CAR molecule. **d** Dependence of the function of the hEGFR/VEGF-specific AvidCAR on the presence of VEGF. The figure shows the lysis of hEGFRt$^{pos}$ Jurkat target cells by T cells expressing the hEGFR/VEGF-specific AvidCAR (flow cytometry-based assay; two independent experiments with three different donors, *$p < 0.05$, calculated by two-tailed paired $t$-test). Mock-T cells (no CAR) or T cells expressing a monovalent high-affinity αhEGFR-Ser-BBz-CAR served as negative and positive controls, respectively. VEGF was added as indicated. AP20187 was added for homodimerization of the target antigen hEGFRt via its integrated FKBP12-36V-domain. Exact $p$ values are given in Supplementary Data 2. Source data are provided as a Source data file.

direct toxic effects caused by VEGF. Summing up, we designed an AvidCAR that recognizes the combination of a cell surface and a soluble antigen, thus further broadening the application range of the AvidCAR platform.

**Regulation and specificity of AvidCARs in vivo.** As shown above, the activity of ON-switch AvidCARs can be regulated by conditional dimerization with small molecules. To test this principle in vivo, we lentivirally transduced primary human T cells with a hEGFR-specific ON-switch AvidCAR, whose antigen-binding domain specifically recognizes human, but not murine EGFR (mEGFR) (Supplementary Fig. 7a). This CAR additionally harbors the FKBP12-F36V domain for conditional homodimerization by AP20187. CD19$^{pos}$ Nalm-6 target cells were stably transduced with hEGFRt (Supplementary Fig. 7b), enabling the direct comparison of the hEGFR-directed AvidCAR with two control CARs: (1) a high-affinity version of this Avid-CAR and (2) a CD19-specific reference BBz-CAR based on the scFv FMC63 (αCD19-BBz; Supplementary Data 1). Initial in vitro experiments confirmed that (1) both control CARs (αCD19-BBz-CAR and high-affinity αhEGFR-Ser-BBz-CAR) efficiently triggered lysis of CD19$^{pos}$/hEGFRt$^{pos}$ Nalm-6 cells, (2) that the activity of the hEGFR-specific ON-switch AvidCAR was dependent on the presence of the dimerizer AP20187 (Fig. 5a) and that (3) the parental hEGFRt$^{neg}$ Nalm-6 cells were not recognized by hEGFR-directed CARs (Fig. 5b). Similar effects were observed irrespective of the effector:target ratio used (Fig. 5a and b). In vivo, both control CARs efficiently delayed tumor outgrowth. Remarkably, the switchable αhEGFR-AvidCAR mediated comparable tumor control, but only upon administration of the dimerizing small molecule (on-state; Fig. 5c, Supplementary Fig. 7c). Moreover, even if the switchable αhEGFR-AvidCAR-T cells were initially administered without the dimerizing small molecule (off-state, leading to tumor outgrowth), the data indicate that antitumor activity could still be triggered by a single administration of the small molecule 11 days after CAR-T cell administration (Fig. 5c and Supplementary Figs. 7c and 8a). To investigate the effects of these CAR-T cells over a longer time span, we generated a Kaplan–Meier curve (Fig. 5d). Again, while the αhEGFR-AvidCAR-T cells had no effect in the absence of the dimerizer, long-term administration of the dimerizing small molecule resulted in significant prolongation of survival—similar to that observed upon treatment with either the high-affinity version of the αhEGFR-CAR or the standard αCD19-BBz-CAR. Together, these results confirm that CAR function can be controlled in vivo by regulating CAR avidity. Importantly, this AvidCAR represents a clinically applicable ON-switch CAR, as the

domain FKBP12-F36V (used for conditional homodimerization by AP20187 in this experiment) can also be homodimerized by the closely related investigational drug AP1903 (rimiducid).

Finally, we also tested the AND-gate function of an AvidCAR in vivo. For this purpose, we inserted domains for constitutive heterodimerization into the hEGFR/hHER2-specific AvidCAR described above, since the persistent administration of sufficient amounts of the currently available heterodimerizer AP21967 in mice is not feasible[21]. That is, the domains for conditional heterodimerization were replaced with drug-independent hetero-dimerizing leucine zipper domains[10,47] (EE and RR, respectively; Fig. 6a; Supplementary Data 1). The two polypeptide chains of this αhEGFR/αhHER2-AvidCAR were expressed at high levels (Supplementary Fig. 8b, c) and formed a functional CAR with potent AND-gate function, as demonstrated by specific lysis of hEGFRt$^{pos}$/hHER2t$^{pos}$ target cells, but no or only weak lysis of single-positive target cells (Fig. 6b). Subsequently, this αhEGFR/αhHER2-AvidCAR was introduced lentivirally into primary human T cells and tested for its function in an in vivo model, where four different Nalm-6 populations (hEGFRt/hHER2t double-positive Nalm-6, both types of single-positive controls and double-negative version) were mixed 1:1:1:1 and subsequently administered intravenously (i.v.) to NSG mice. Ten days after CAR-T cell inoculation, the four different Nalm-6 populations were analyzed in the bone marrow, liver, blood, and spleen (Fig. 6c, d, and Supplementary Figs. 9–11). Strikingly, the αhEGFR/αhHER2-AvidCAR-T cells specifically killed double-positive, while ignoring single-positive Nalm-6 cells, even though they were present in the same tissue of the same animal (Fig. 6c, d and Supplementary Fig. 10). Notably, this high specificity was obtained despite even higher antigen expression levels in the single-positive controls (Fig. 6c and Supplementary Fig. 8d). In addition to its high specificity, the efficacy of the αhEGFR/αhHER2-AvidCAR-T cells was comparable to that of high-affinity αhEGFR-CAR-T cells (equally efficient in the liver; only slightly less efficient in the bone marrow; Fig. 6d and Supplementary Fig. 10). No Nalm-6 cells were detected in the blood and spleen of diseased mice (Supplementary Fig. 11). As expected, the high-affinity αhEGFR-CAR mediated killing of both hEGFRt/hHER2t double-positive and hEGFRt single-positive Nalm-6 cells (Fig. 6c, d and Supplementary Fig. 10). Together, these in vivo data demonstrate that the αhEGFR/αhHER2-AvidCAR is both efficacious and highly specific, even if double-positive target cells and single-positive controls are co-localized in the same tissues. To the best of our knowledge, this is the first example of an AND-gate CAR that can selectively target double-positive cells without requiring a safe distance between single-positive healthy bystander cells.

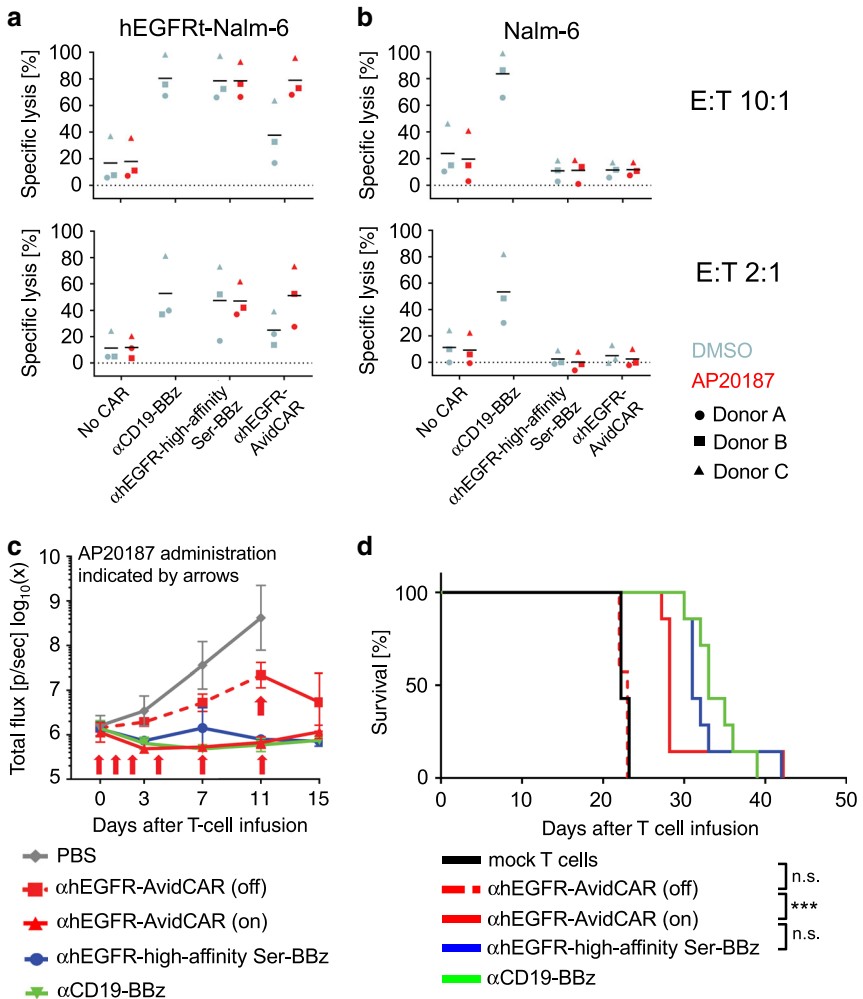

**Fig. 5 Regulation of an ON-switch AvidCAR in vivo. a, b** In vitro characterization of the lentivirally transduced CAR-T cells and hEGFRt^pos Nalm-6 target cells subsequently used in vivo. The hEGFR-specific CARs αhEGFR-high-affinity Ser-BBz and αhEGFR-AvidCAR contained either the high- or low-affinity rcSso7d binder, respectively, fused to a Ser-BBz backbone with an integrated FKBP12-F36V-domain for CAR homodimerization by AP20187. A CD19-specific CAR (αCD19-BBz) comprising a high-affinity scFv (FMC63) fused to a BBz backbone was included as a benchmark (SEQ-ID 19). The diagrams show the cytolytic activity of mock-T cells (no CAR) and different CAR-T cells in co-cultures with Nalm-6 cells ± hEGFRt expression (Luciferase-based assay; E:T ratio 10:1 or 2:1, as indicated; one experiment with three different donors). AP20187 was added where indicated. **c** Bioluminescence imaging of Nalm-6 cells over time in NSG mice. Three days after intravenous injection of $0.5 \times 10^6$ hEGFRt-Nalm-6 cells, mice were intravenously injected with $10 \times 10^6$ CAR-T cells or PBS (one experiment with five mice per group, in the αEGFR-high-affinity Ser-BBz group one mouse died 1 day after tumor injection). Mice which received the αhEGFR-AvidCAR-T cells were intraperitoneally injected with AP20187 (indicated by arrows) either starting from the day of T-cell injection (αhEGFR-AvidCAR on) or only once on day 11 thereafter (αhEGFR-AvidCAR off). Data are presented as geometric means ± geometric SD. **d** ON-switch AvidCAR-T cells prolong survival. NSG mice were injected via tail vein with $0.5 \times 10^6$ hEGFRt-Nalm-6 cells and 3 days later with $10 \times 10^6$ CAR-T cells or mock-T cells, as indicated. Mice were intraperitoneally injected three times a week starting from the day of T-cell injection either with 2 mg/kg AP20187 (αhEGFR-AvidCAR group) or vehicle solution (all other groups). Kaplan–Meier plot shows the survival of seven mice per group (\*\*\*$p < 0.001$, calculated by a two-sided log-rank analysis). Exact $p$ values are given in Supplementary Data 2. Source data are provided as a Source data file.

## Discussion

In this study we present the concept of AvidCARs. The key design principles were (1) the development of strictly monomeric CAR backbones by removing cysteines in the hinge region to prevent uncontrolled disulfide-mediated CAR dimerization; (2) the incorporation of antigen-binding domains whose affinities are substantially reduced down to levels comparable to those of TCRs ($K_d > 0.5 \mu M$)[48]; and (3) the use of single-domain antigen-binding domains, which seems to be preferable to exclude potential oligomerization that has been observed for some scFvs.

As described above, AvidCARs were successfully engineered using an EGFR-targeting rcSso7d-based binder and an affibody directed against HER2. Of note, similar dimerization-dependent AvidCAR activation was also observed when using low-affinity

nanobodies targeting green fluorescent protein (GFP)[49] and HER2[50], (Supplementary Fig. 12). Thus, in total we have successfully generated AvidCARs with four different targeting domains based on three different protein scaffolds, demonstrating the generalizability of the AvidCAR concept. An important feature, that all of these targeting domains (rcSso7d, affibodies and nanobodies) have in common, is their single-domain architecture. In contrast, scFvs are composed of two domains, which are known to cause mispairing and thus oligomerization to various extent depending on the scFv mutant, linker length, and environmental conditions[51,52]. This has long been known for soluble scFvs[52] and recently such scFv clustering has also been associated with tonic signaling in CAR-T cells[34]. In our study, we provide direct biochemical evidence that scFv clustering indeed takes

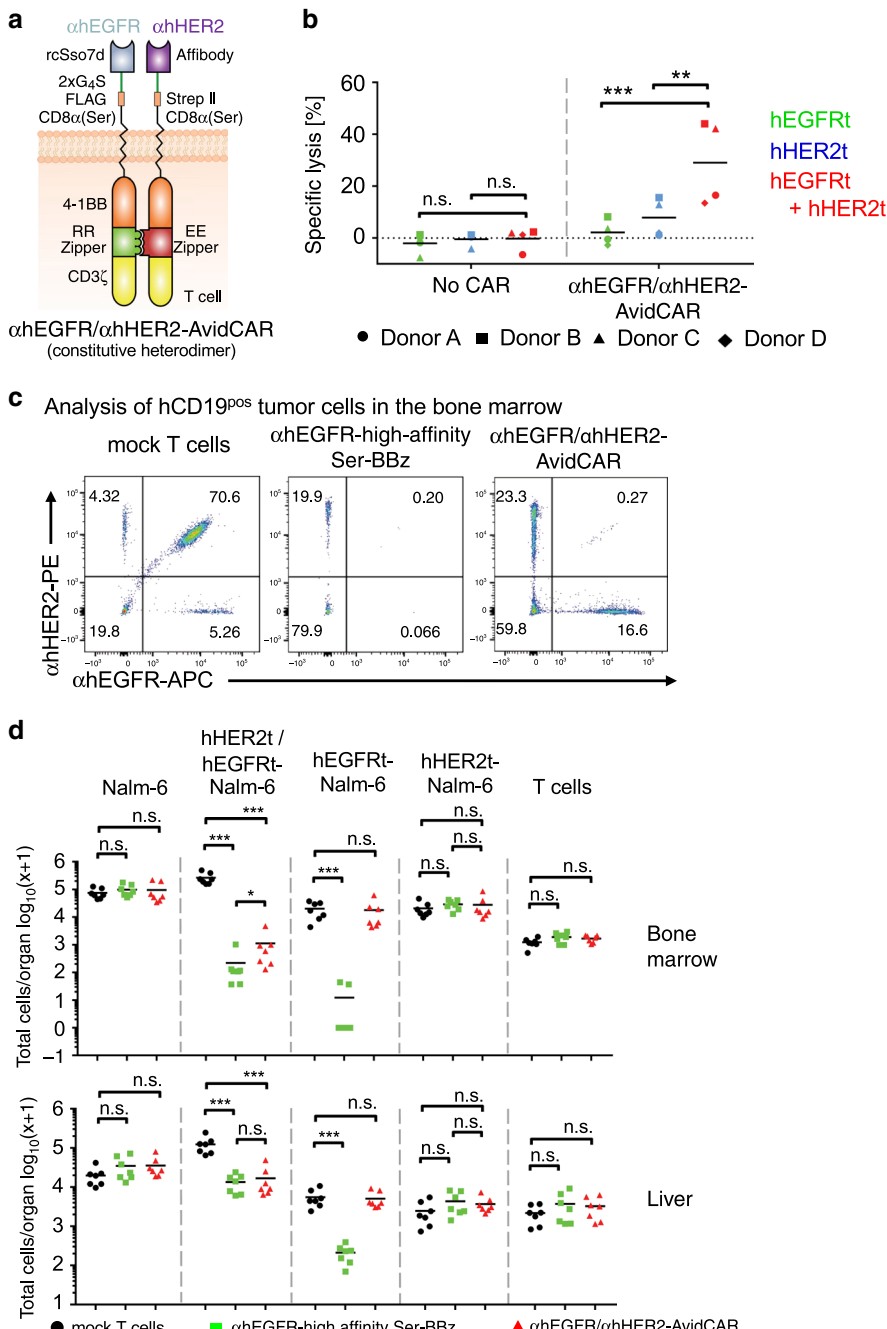

**Fig. 6 Specificity of an AND-gate AvidCAR in vivo. a** Schematic of a constitutively heterodimeric leucine zipper based AND-gate AvidCAR (αhEGFR/αhHER2-AvidCAR; SEQ-IDs 25 and 26). **b** Lysis of single- and double-antigen positive target cells by T cells expressing either no CAR or the αhEGFR/αhHER2-AvidCAR (flow cytometry-based assay; three independent experiments with four different donors; ***$p$ < 0.001, **$p$ < 0.01, calculated by a mixed model, with CAR group, antigen, and their interaction as fixed effects and the donor as a random effect to take into account the correlated structure, unadjusted two-sided $p$ values are given). T cells were co-cultured with mRNA-electroporated Jurkat target cells expressing either hEGFRt or hHER2t or both antigens. **c**, **d** Specificity and efficacy of AND-gated AvidCAR-T cells in vivo. NSG mice were injected via tail vein with 0.5 × 10⁶ cells of a 1:1:1:1 mixture of Nalm-6, hEGFRt-Nalm-6, hHER2t-Nalm-6, and hHER2t-hEGFRt-Nalm-6 cells (Supplementary Fig. 9a) and 3 days later with 10 × 10⁶ CAR-T cells or mock-T cells, as indicated. As a positive control we used the high-affinity αhEGFR-CAR, i.e., the Ser-BBz backbone (with integrated FKBP12-F36V domain) fused to the high-affinity EGFR-binder, from which the low-affinity version for the AvidCAR was derived. Mice were sacrificed 13 days after tumor injection. Bone marrow and liver was analyzed for tumor cell infiltration. **c** Representative flow cytometric analysis of the ratio of the four different Nalm-6 cell populations in bone marrow. See Supplementary Fig. 10 for the flow cytometric analyses of the organs of all mice and Supplementary Fig. 9b for the used gating strategy. **d** Total number of Nalm-6 cells and T cells in the bone marrow and liver of NSG mice ($n$ = 7, every data point represents one animal; *$p$ < 0.05, ***$p$ < 0.001, calculated by a two-way ANOVA analysis of log-transformed values). Exact $p$ values are given in Supplementary Data 2. Source data are provided as a Source data file.

place on the CAR-T cell surface. Moreover, our experiments also demonstrated that scFvs that tend to cluster do not allow for the generation of efficient AvidCARs. However, we do not exclude that other scFvs may be suitable for the construction of Avid-CARs. We would also like to note that we do not propose the separate expression of VH and VL as a suitable solution in case of oligomerizing scFvs. This design was only used as a model system to study the mechanism of scFv clustering (Fig. 3d, e). Instead, we strongly favor the use of single-domain binding scaffolds, especially in the context of AvidCARs. In fact, various different single-domain binding scaffolds have already been used in CARs and due to their favorable properties might be attractive for CARs in general. Regarding the potential immunogenicity it should be noted that while the single-domain binding scaffolds used here are based on nonhuman proteins, there are also several alternatives derived from human protein domains, such as monobodies.

Our experiments demonstrate that AvidCARs with low-affinity antigen-binding domains require bivalent interaction to be efficiently triggered. Such behavior can be explained by the fact that CAR signaling, i.e., the sufficient phosphorylation of signaling proteins, requires a certain duration of interaction with its target antigens, as has been shown for the TCR and other immunoreceptors[39,53–55]. For low-affinity binding domains, this duration is too short in a monovalent interaction and must be prolonged by interaction via a second binding domain, as supported by both our mathematical model and experimental data, in particular those presented in Fig. 2e. We demonstrate that this mechanism can be exploited to regulate CAR function by conditional dimerization and/or for combinatorial recognition of two different antigens. Such AND-gate function mediated by bivalent interaction of a bispecific CAR would not be possible with current homodimeric CAR formats, which may explain why bispecific CARs have so far only been used as OR gates[42–44]. Finally, due to the dependence on bivalent interaction, we could also use our AvidCAR platform as a functional reporter system for identifying clustering tendencies of different CAR components such as a hHER2-specific scFv.

The AND-gate AvidCAR presented in Fig. 4a, b combines both small molecule-mediated regulation (ON-switch function) and combinatorial antigen recognition (AND-gate function). However, since the required small molecule AP21967 is not suited for long term in vivo experiments[21], we replaced this AP21967-responsive heterodimerization module with a constitutive version based on leucine zippers to be able to study this AND-gate AvidCAR in vivo (Fig. 6). Of note, recently we introduced a small molecule-regulated heterodimerization system, which is based on human retinol binding protein 4, an engineered protein scaffold and the orally available small-molecule drug A1120[29]. Thus, this molecular ON-switch represents an alternative to the AP21967-based system, potentially enabling the future design of AND-gate AvidCARs that can additionally be controlled with a small-molecule drug shown to be nontoxic in mice, even when administered at high doses up to 30 mg/kg for several months[56].

A crucial feature of the avidity-based AND-gate function of AvidCARs is the fact that antigens A and B must be co-expressed on the same cell. This is in contrast to SynNotch CAR-T cells, in which CAR expression can be induced by healthy cells expressing antigen A, which subsequently enables the recognition of nearby healthy cells expressing antigen B[30]. Another potential advantage of AvidCAR-T cells is that they are also fully functional when target cells are rare, as AvidCAR expression—unlike CAR expression in SynNotch T cells—does not need to be induced by antigen-expressing cells. Finally, the independence of AvidCARs from a xenogeneic transcription factor reduces the risk of immunogenicity.

Another consequence of the avidity-based mechanism of AvidCARs is the strong capability to sense antigen density. Antigen density determines the probability that a second binding domain can catch an antigen before the first one dissociates from its antigen. That is, antigen density strongly determines the lifetime of the interaction between a bivalent low-affinity CAR and its antigens—and thus CAR signaling. This mechanism is exploited in affinity tuning of current CARs to limit CAR-T cell cytotoxicity to tumor cells expressing elevated target antigen levels[15–20]. Due to the same mechanism, AND-gate AvidCARs integrate the dependence on two different antigens (i.e., AND-gate function) with the ability to sense antigen density, which can further increase tumor specificity.

Nevertheless, for other applications, it is desirable to enhance the antigen sensitivity of the AvidCARs, e.g., for antigens expressed at low levels and/or to reduce the risk of tumor escape due to antigen downregulation. It is expected that avidity effects—and thereby CAR sensitivity—can be further increased by optimizing the design of AvidCARs (e.g., length and flexibility of spacers), which was not investigated in the examples presented here. However, we show that antigen dimerization enhances the avidity effect and thereby AvidCAR sensitivity. Therefore, co-localized antigens are particularly attractive candidates. Of note, many TAAs naturally exist in a dimeric/oligomeric state[57–59], are clustered into nanodomains or interact directly with other antigens. Prominent examples of co-localization include heterodimers of EGFR family members either among themselves or with other growth factor receptors, or the co-localization of receptors and gangliosides[60–63]. The architecture of the mammalian cell membrane is still incompletely understood and many more co-localized proteins may exist that are not known yet[64,65].

Together, our study highlights and explores a previously neglected aspect in the CAR field, which is the bivalent antigen engagement due to the dimeric nature of virtually all currently applied CAR molecules. Therefore, the presented data will contribute to an improved mechanistic understanding of CAR function and biology. Moreover, we present a strategy for specifically exploiting this bivalent antigen recognition in a controlled manner, ultimately opening up a range of applications in which an AvidCAR integrates multiple input signals, e.g., (1) a cell surface antigen and a clinically applied small molecule, (2) two different cell surface antigens and a small molecule or (3) a cell surface antigen and a soluble factor enriched in the tumor microenvironment. Thus, we expect that AvidCARs will be a highly valuable platform for the development of controllable CAR therapies with improved tumor specificity.

## Methods

**Cell culture**. Buffy coats from de-identified healthy donor's blood were purchased from the Austrian Red Cross, Vienna, Austria. Primary human T cells were isolated by negative selection using the RosetteSep Human T-Cell Enrichment Cocktail (STEMCELL Technologies) and cryopreserved in RPMI-1640 GlutaMAX medium (Thermo Scientific) supplemented with 20% FCS (Sigma-Aldrich) and 10% DMSO (Sigma-Aldrich) until further use. After thawing, primary human T cells were immediately activated with Dynabeads Human T-Activator αCD3/αCD28 beads (Thermo Scientific) at a 1:1 ratio according to the manufacturer's instructions. T cells were expanded before the experiments for at least 14 days in T-cell medium, consisting of RPMI-1640 GlutaMAX supplemented with 10% FCS, 1% penicillin–streptomycin (Thermo Scientific) and 200 U/mL recombinant human IL-2 (Peprotech). Half of the medium was exchanged every other day and cell densities were kept between 0.4 and $2 \times 10^6$ cells/mL. Nalm-6 cells and Jurkat cells (gifts from Dr. Sabine Strehl and Dr. Michael Dworzak, respectively; CCRI, Vienna, Austria) were maintained in RPMI-1640 GlutaMAX supplemented with 10% FCS and 1% penicillin–streptomycin. Lenti-X 293T cells (Takara) were maintained in DMEM (Thermo Scientific) supplemented with 10% FCS. Nalm-6 target cell lines ±hEGFRt and/or ±hHER2t expression were established by transduction with a lentiviral vector encoding firefly luciferase (ffLuc), puromycin N-acetyl transferase and hEGFRt and/or hHER2t. A stable clone of Nalm-6 cells expressing both transgenes was established by limiting dilution cloning and subsequently was co-transduced with a lentiviral vector encoding ffLuc and enhanced GFP (eGFP) to

ensure high expression of the luciferase reporter gene. Cell lines were regularly tested for mycoplasma contamination using the MycoAlert PLUS Mycoplasma Detection Kit (Lonza) and for squirrel monkey retrovirus contamination by PCR (performed by Labdia Labordiagnostik GmbH, Austria). Cell line authentication was performed by SNP-profiling at Multiplexion GmbH, Germany.

The homo- and heterodimerization of proteins in vitro was induced by the addition of AP20187 or AP21967 (both Takara) to the cell suspensions at a final concentration of 10 and 500 nM, respectively. The effective concentrations of both compounds were determined by the addition of increasing concentrations to Jurkat cells expressing hEGFRt, which could be homo- or heterodimerized (Supplementary Fig. 4d).

**In vitro transcription and electroporation of mRNA**. DNA encoding the respective transgene was amplified by PCR and subsequently transcribed in vitro with the mMessage mMachine T7 Ultra Kit (Ambion) according to the manufacturer's instructions. The resulting mRNA was purified using the RNeasy Kit (Qiagen). Electroporation was performed with $<1 \times 10^7$ cells/100 μl Opti-MEM (without phenol red; Thermo Scientific) using a Square wave protocol (1 pulse, 500 V) on the Gene Pulser Xcell Electroporation System (Biorad) and 4 mm electroporation cuvettes (VWR). The length of the pulse was 5 ms for primary human T cells and 3 ms for Jurkat cells. Primary human T cells were electroporated with 5 μg of CAR-mRNA and Jurkat cells were electroporated with different mRNA amounts as indicated. After electroporation, the cells were immediately recovered in warm cell growth medium and incubated over night at 37 °C.

**Lentiviral transduction**. To generate VSV-G pseudotyped lentivirus, Lenti-X 293T cells (Takara) were co-transfected with a third-generation puromycin-selectable pCDH expression vector (System Biosciences) and second-generation viral packaging plasmids pMD2.G and psPAX2 (Addgene plasmids #12259 and #12260, respectively; gifts from Didier Trono) using the PureFection Transfection Reagent (System Biosciences) according to the manufacturer's instructions. Supernatants were collected on day 2 and 3 after transfection and were concentrated 100-fold using the Lenti-X Concentrator (Takara) according to the instructions provided by the manufacturer. Viral suspensions were resuspended in RPMI-1640 medium supplemented with 10% FCS and 1% penicillin–streptomycin and frozen at −80 °C. Twenty-four hours after their activation with αCD3/αCD28 beads (Thermo Scientific), primary human T cells were transduced in cell culture plates, which were coated according to the manufacturer's instructions with Retronectin (Takara). Thawed viral supernatant was added to the T cells ($0.5 \times 10^6$ cells/mL) at a final dilution of 1:2. To ensure high and uniform expression of the transgenes, 1 μg/mL puromycin (Sigma-Aldrich) was added 3 days later and removed after 2 more days. Transduced T cells were expanded in AIMV medium (Thermo Scientific) supplemented with 2% Octaplas (Octapharma), 1% L-Glutamine (Thermo Scientific), 2.5% HEPES (Thermo Scientific), and 200 U/mL recombinant human IL-2. Half of the medium was exchanged every 2–3 days. Cell lines were transduced by exposure to varying concentrations of lentiviral supernatants for 3 days, followed by puromycin selection for 2 days with varying concentrations of puromycin and expansion in the respective growth medium.

**Flow cytometric analysis**. Cells were counted using AccuCheck Counting Beads (Thermo Scientific). Propidium iodide (Sigma-Aldrich) was added for exclusion of dead cells. Unspecific antibody binding was prevented by preincubation of the cells for 10 min at 4 °C in FACS buffer [i.e., PBS (Thermo Scientific), 0.2% human albumin (CSL Behring) and 0.02% sodium azide (Merck KGaA)] supplemented with 10% (v/v) human serum. Alternatively, for subsequent detection of Fc-containing CARs, preincubation was done in antihuman FACS buffer (i.e., PBS, 10% FCS and 0.02% sodium azide) supplemented with 10% (v/v) murine IgG1-κ (clone MOPC21; 1 mg/mL; Sigma-Aldrich). This preincubation was omitted for the detection of the FMC63-derived αCD19-CAR by the protein L-biotin conjugate (GenScript; dilution 1:167). The following antibodies were used: αFLAG tag (PE; clone L5; BioLegend; dilution 1:167), αFLAG tag (APC; clone L5; BioLegend; dilution 1:167), αStrep II tag (biotin; clone 5A9F9; GenScript; dilution 1:500), αhEGFR (APC and PE; clone AY13; BioLegend; dilution 1:50), and αhHER2 (PE; clone 24D2; BioLegend; dilution 1:50), αhCD19 (BV421; clone HIB19, BioLegend; dilution 1:167), αhCD45 (PerCP; clone 2D1; BD Biosciences; dilution 1:50), αhCD3 (PE-Cy7; clone SK7; BD Biosciences; dilution 1:50), αGFP (PE; clone FM264G; BioLegend; dilution 1:50). Binding of hEGFR-Fc and mEGFR-Fc (both R&D) to hEGFR-specific CARs was detected by an antibody directed against human IgG1-Fc (PE; clone JDC-10; Southern Biotech; dilution 1:25). Cells were incubated with the respective antibodies or protein L for 25 min at 4 °C and then washed twice with ice-cold FACS buffer or—if Fc-containing CARs were detected—antihuman FACS buffer. Biotinylated antibodies and protein L were further incubated with streptavidin APC or streptavidin PE (both Thermo Scientific; dilution 1:250) for 25 min at 4 °C and then again washed twice. For the analysis of single-cell suspensions of mouse organs, dead cells were excluded by staining with the Fixable Viability Dye eFluor™ 780 (Thermo Scientific). Subsequently, the cells were acquired on an LSR Fortessa instrument (BD Biosciences) and analyzed using the FlowJo Software (Version 10.6.1; FlowJo, LLC) and the BD FACSDiva™ Software V8.0.1. Nontransfected cells or isotype antibodies served as negative controls.

**Quantitation of target antigen density**. The surface density of target antigen molecules on Jurkat cells was quantified using the QuantiBRITE Phycoerythrin (PE) Fluorescence Quantitation Kit (Becton Dickinson) according to the manufacturer's instructions. Briefly, the cells were stained with a saturating concentration of a PE-labeled αhEGFR-antibody (clone AY13; BioLegend). Subsequently, the geometric mean of the fluorescence intensity was determined, subjected to background subtraction using unstained or isotype-labeled cells and then used to estimate the number of antibodies bound per cell (ABC). ABC values were corrected for the PE-conjugation efficiency of the respective antibody (i.e., PE dye-to-antibody ratio) yielding effective surface densities.

**Generation of mutant binders by alanine scanning**. Site-directed mutagenesis of the hEGFR- and hHER2-specific binder scaffolds was performed using the Quik-Change Lightning Site-Directed Mutagenesis Kit (Agilent Genomics), according to the manufacturer's instructions. Mutagenic primers were designed using the QuikChange Primer Design Software (Agilent Genomics) and synthesized by Biomers, Germany. Primer sequences can be found in Supplementary Data 3. In the case of the hHER2-specific binder, the affibody zHER2-AK was generated from the parental affibody zHER2:4, which binds to hHER2 with an affinity of 50 nM[45], prior to the alanine scan by introducing mutations N23A and S33K for preventing N-glycosylation[66].

**Determination of binding affinities on cell membranes**. Jurkat cells with high level expression of hEGFRt were obtained by electroporation with 3 μg of hEGFRt-mRNA. On the next day, the cells were washed with PBS, resuspended in PBSA [PBS supplemented with 0.1% BSA (Sigma-Aldrich)] and incubated in 96-well plates for 1 h at 4 °C with varying concentrations of fluorescent chimeric hEGFR-binder proteins (i.e., E11.4.1 or mutants thereof fused to sfGFP). The plates were then centrifuged ($450 \times g$, 7 min, 4 °C), the supernatant discarded and the cells resuspended in ice-cold PBSA immediately prior to acquisition on an LSR Fortessa instrument. The cells were kept on ice during the analysis to avoid endocytosis. $K_d$ values were calculated by curve fitting using Microsoft Excel (Microsoft Corporation).

**Construction of transgenes**. A detailed description of all used transgenes is given in Supplementary Data 1. hEGFRt and hHER2t were generated using the Addgene plasmids #11011[67] and #16257[68], respectively (gifts from Matthew Meyerson and Mien-Chie Hung). sfGFP was derived from the Addgene plasmid #54737[69] (gift from Michael Davidson & Geoffrey Waldo). Cytoplasmic domains of CD28, ICOS, OX40, and CD2 were derived from cDNA clones (Sino Biological). The cytoplasmic domain of CD3ζ contained a Q65K mutation (Uniprot P20963-3). Synthetic DNAs were synthesized by GeneArt (Thermo Scientific). Assembly of DNA molecules into complete constructs was performed using the Gibson Assembly Master Mix (New England BioLabs) according to the manufacturer's instructions.

**Recombinant expression of binder proteins**. To determine their affinities, the binders were expressed recombinantly as soluble proteins with an N-terminal hexahistidine and a C-terminal sfGFP fusion tag using the pE-SUMO vector (Life Sensors), in which the SUMO-tag was deleted. Briefly, *Escherichia coli* cells (Tuner DE3) were transformed with respective plasmids using heat shock transformation. After overnight incubation in lysogeny broth (LB) medium at 37 °C, cultures were diluted 1:100 in terrific broth medium supplemented with kanamycin (50 μg/mL) and were kept shaking at 37 °C. Expression of the transgene was started when cultures reached an A600 of 2 by addition of 1 mM of isopropyl β-D-1-thiogalactopyranoside (IPTG) followed by overnight incubation at 20 °C. Transformed cells were harvested by centrifugation ($5,000 \times g$, 20 min, 4 °C) and resuspended in sonication buffer (50 mM sodium phosphate, 300 mM NaCl, 3% glycerol, 1% Triton X-100, pH 8.0). Cell disruption was accomplished by sonication ($2 \times 90$ s, duty cycle 50%, amplitude set to 5) followed by removal of cell debris by centrifugation ($20,000 \times g$, 30 min, 4 °C). Hexahistidine-tagged proteins were extracted from crude cell lysates by immobilized metal affinity chromatography using TALON metal affinity resin (Takara). Sonicated supernatants were supplemented with 10 mM imidazole and applied onto the resin twice, followed by repeated washing steps with equilibration buffer (50 mM sodium phosphate, 300 mM NaCl, pH 8.0) supplemented with increasing concentrations of imidazole (5–15 mM). Purified proteins were eluted by washing the column with equilibration buffer with 250 mM imidazole. Buffer exchange to PBS was performed with Amicon Ultra-15 10 K centrifugal filters (Merck Millipore). Affibody-based binder scaffolds were further purified by performing a preparative SEC using a Superdex 200 column (10 mm × 300 mm, GE Healthcare). Purified proteins were directly frozen at −80 °C. Truncated human VEGF (Uniprot P15692 amino acids 40–134) was expressed as previously described[46]. Briefly, *E. coli* BL21 (DE3) cells were transformed with the pJ414 vector (ATUM) including the gene encoding for truncated VEGF. LB medium supplemented with ampicillin (100 μg/mL) was inoculated with an overnight culture. Once the culture reached an A600 of 2 at 37 °C, the overexpression was induced by addition of 1 mM IPTG. After 4 h of shaking at 37 °C, cells were harvested by centrifugation ($5,000 \times g$, 20 min, 4 °C). The pellet was resuspended in 20 mM TRIS-HCl buffer (pH 7.5) including 5 mM ethylenediaminetetraacetic acid (EDTA), incubated for 10 min and disrupted by

ultrasonication with a Vibra-Cell 375 ultrasonic processor (Sonics & Materials, Inc.). After two washing steps using the same buffer, the pellet was resuspended in 20 mL unfolding buffer (20 mM TRIS-HCl, 5 mM EDTA, 7.5 M urea, 4 mM dithiothreitol, pH 7.5), stirred for 2 h at room temperature followed by centrifugation ($39,000 \times g$, 25 min, 4 °C). The supernatant containing the unfolded protein was diluted tenfold into refolding buffer (20 mM TRIS-HCl, 7 mM $CuCl_2$, pH 8.4) and stirred overnight at room temperature. Following a dialysis step against 20 mM TRIS-HCl (pH 8.0), the protein was purified using a 6 mL Resource Q column, a 1 mL HiTrap Phenyl FF (Low Sub) column and a HiLoad 16/600 Superdex 200 pg (all from GE Healthcare) according to the manufacturer's instructions. Pure VEGF samples were stored at −80 °C.

**Size exclusion chromatography.** For analytical SEC analysis, recombinant proteins were diluted in SEC running buffer (PBS supplemented with 200 mM NaCl) and filtered through a 0.1 μm Ultrafree MC VV centrifugal filter (Merck Millipore). Subsequently, 25 μg protein was loaded onto a Superdex 200 column (10 mm × 300 mm, GE Healthcare) connected to an HPLC Prominence LC20 System (Shimadzu) at a flow rate of 0.75 mL/min at 25 °C.

**Surface plasmon resonance.** SPR analysis was performed with a Biacore T200 instrument (GE Healthcare). All experiments were performed in degassed and filtered PBS, pH 7.4, supplemented with 0.1% BSA (Sigma-Aldrich) and 0.05% Tween-20 (Merck Millipore) at 25 °C. Immobilization of hEGFR-Fc and hHER2-Fc (both R&D) was performed on a Protein A sensor chip (GE Healthcare) at a flow rate of 10 μL/min for 60 s at a concentration of 6.67 and 4 μg/mL, respectively, yielding a density of ~400 response units. Five different concentrations of the respective binder scaffold (depending on the expected $K_d$ value) were injected at a flow rate of 30 μL/min for 15 s (for binders E11.4.1, E11.4.1-G25A, E11.4.1-G32A, and zHER2-AK-R10A) or 60 s (for zHER2-AK) in the single-cycle kinetic mode. Subsequently, a dissociation step was performed for 30 s (all E11.4.1-based binders), 60 s (zHER2-AK-R10A), or 180 s (zHER2-AK). Protein A sensor chips were regenerated by applying 10 mM Glycine-HCl, pH 1.5, at a flow rate of 30 μL/min for 30 s. The $K_d$ value was either obtained by steady state binding analysis or by fitting the sensorgram to a 1:1 Langmuir model in the kinetic mode using the Biacore T200 Evaluation Software (GE Healthcare).

**Western blotting and crosslinking of membrane proteins.** Crosslinking of membrane proteins was performed with disuccinimidyl suberate (DSS) (Thermo Scientific) according to the manufacturer's instructions. Briefly, Jurkat cells were washed three times with ice-cold PBS. Cells were resuspended in PBS to a cell concentration of $20 \times 10^6$ cells/mL. Subsequently, DSS was added at a concentration of 5 mM and the reaction was incubated for 30 min at room temperature. Chemical crosslinking was stopped by addition of the Quench Solution (1 M TRIS, pH 7.5), yielding a final concentration of 20 mM TRIS. Cells were washed once in PBS and lysed in ice-cold RIPA buffer (Sigma-Aldrich), supplemented with protease inhibitors (Halt Protease Inhibitor Cocktail; Thermo Scientific) and nucleases (Pierce Universal Nuclease; Thermo Scientific). The lysate was incubated on ice for 5 min. Subsequently, samples were denatured for 5 min at 50 °C in loading buffer (LDS Sample Buffer; Thermo Scientific) and frozen at −20 °C. Cleared lysates ($14,000 \times g$, 10 min, 4 °C) were resolved on 4–20% gradient SDS-PAGE gels (Mini-PROTEAN TGX Precast Protein Gels; BioRad) under nonreducing conditions. Separated proteins were transferred to 0.45 μm nitrocellulose membrane (GE Healthcare) and blocking was performed with Western Blocking Reagent (Roche) for 1 h at room temperature. Membranes were probed with an antibody against hEGFR (clone 528; 1:500 dilution; Thermo Scientific) and incubated overnight at 4 °C. Subsequently, membranes were incubated with a secondary goat antimouse horseradish peroxidase-conjugated antibody (Thermo Scientific) for 1 h at room temperature and bands were developed using the SuperSignal West Femto Maximum Sensitivity Substrate (Thermo Scientific), according to the manufacturer's instructions. Loading control was performed by probing the membrane with an antibody against GAPDH (clone ab9485; 1:5,000; Abcam) for 1 h at room temperature. Detection was performed by incubating the membrane with a secondary goat antirabbit DyLight 800-conjugated antibody (Thermo Scientific) for 1 h at room temperature and visualizing the bands on the Odyssey Imaging System with the Odyssey Infrared Imaging System software (Version 3.0, LI-COR).

**Cytotoxicity assay.** Cytotoxicity of primary human T cells was assayed with either a luciferase- or a flow cytometry-based assay. Luciferase-based assay: 10,000 target cells (i.e., luciferase-expressing Nalm-6 or Jurkat cells) were co-cultured for 4 h at 37 °C with 20,000 primary human T cells (= E:T cell ratio 2:1, unless stated otherwise) in white round-bottom 96-well plates (Sigma-Aldrich) in 100 μL RPMI medium without phenol red (Thermo Scientific) supplemented with 10% FCS and 1% penicillin–streptomycin. After co-culture, viable target cells were quantified by determining the luciferase activity. Hereto, the culture plates were equilibrated for 10 min to room temperature and subsequently luciferin (Xenolight D-luciferin; Perkin Elmer) was added to the cell suspension (150 μg/mL final concentration). Bioluminescence was quantified after 20 min of incubation at room temperature on an ENSPIRE Multimode plate reader (Perkin Elmer). Specific lysis was calculated

with the following formula:

$$\% \text{ specific lysis} = 100 - \left( \frac{\frac{\text{RLU antigen}^{\text{pos}}\text{target} + \text{effector cells}}{\text{RLU antigen}^{\text{pos}}\text{target cells only}}}{\frac{\text{RLU antigen}^{\text{neg}}\text{target} + \text{effector cells}}{\text{RLU antigen}^{\text{neg}}\text{target cells only}}} \right) \times 100. \quad (1)$$

Specific lysis in Fig. 5a, b was calculated with the following formula:

$$\% \text{ specific lysis} = 100 - \left( \frac{\text{RLU target} + \text{effector cells}}{\text{RLU target cells only}} \right) \times 100. \quad (2)$$

Flow cytometry-based cytotoxicity assay: the assay principle is based on the determination of the ratio of antigen$^{\text{pos}}$ and antigen$^{\text{neg}}$ target cells that expressed either a green or red fluorescent protein, respectively, and were mixed 1:1 before co-culture with T cells. For this purpose, Jurkat cells were electroporated on the day before the assay (1) with mRNA encoding eGFP and mRNA encoding the respective target antigen, and alternatively (2) with mRNA encoding mCherry. Those cells were co-cultured for 4 h at 37 °C in round-bottom 96-well plates with 40,000 primary human T cells at an E:T cell ratio of 4:1:1 (i.e., 40,000 CAR-T cells plus 10,000 antigen$^{\text{pos}}$ and 10,000 antigen$^{\text{neg}}$ Jurkat cells per well). Target cells not co-cultured with CAR-T cells served as a negative control (targets only). After co-culture, the ratio of eGFP- to mCherry-expressing target cells was determined by flow cytometric analysis. Specific lysis was calculated with the following formula:

$$\% \text{ specific lysis} = 100 - \left( \frac{\frac{\% \text{ eGFP}^{\text{pos}} \text{ cells of the sample}}{\% \text{ mCherry}^{\text{pos}} \text{ cells of the sample}}}{\frac{\% \text{ eGFP}^{\text{pos}} \text{ cells of "targets only" control}}{\% \text{ mCherry}^{\text{pos}} \text{ cells of "targets only" control}}} \right) \times 100. \quad (3)$$

**Cytokine secretion by CAR-T cells.** Cytokine secretion of CAR-T cells was determined by co-cultivation with target cells at an E:T cell ratio of 2:1 in flat-bottom 96-well plates for 4 h at 37 °C. Alternatively, cytokine secretion was quantified in the supernatants from cytotoxicity experiments, described above. Supernatants were centrifuged ($450 \times g$, 7 min, 4 °C) and stored at −80 °C. IFN-γ was quantified by enzyme-linked immunosorbent assay (ELISA) using the Human IFN-γ ELISA Ready-SET-Go! Kit (eBioscience/Thermo Scientific) according to the manufacturer's instructions. Analysis was performed on the ENSPIRE Multimode plate reader (Perkin Elmer).

**In vivo experiments.** NOD.Cg-Prkdc$^{\text{scid}}$ Il2rg$^{\text{tm1WJI}}$/SzJ (NSG, The Jackson Laboratory) mice were bred in the Anna Spiegel facility for animal breeding (Vienna, Austria) under standardized conditions (room temperature 22 ± 2 °C, humidity 55 ± 10%, air change rate of 15 and a dark/light cycle of 12 h). Subsequently, mice were transferred to the Preclinical Imaging Laboratory (PIL) or the Center for Biomedical Research of the Medical University of Vienna. All procedures were approved by the Magistratsabteilung 58, Vienna, Austria (GZ: 319093/2014/16) and the Federal Ministry Republic of Austria for Education, Science, and Research (BMBWF-66.009/0243-V/3b/2019). Primary human T cells were lentivirally transduced for CAR expression and expanded for 15–17 days to generate sufficient cell numbers. CAR-T cells directed against hEGFR showed no cross-reactivity with mEGFR as confirmed with recombinant mEGFR-Fc (R&D) (Supplementary Fig. 7a). Nalm-6 cells were engineered to express high levels of the hEGFRt, hHER2t, or both (intracellularly fused to the wild-type FKBP12 domain) and ffLuc [hEGFRt-Nalm-6 (SEQ-ID 20), hHER2t-Nalm-6 (SEQ-ID 23), and hHER2t-hEGFRt-Nalm-6 (SEQ-ID 20 and 23), respectively]. After filtering through a 35 μm cell strainer (Corning Falcon), $0.5 \times 10^6$ of those Nalm-6 cells (in 100 μL PBS) were injected i.v. into the tail veins of NSG mice (male and female, 8–20 weeks of age). Three or 5 days later (as indicated), when tumor burden became detectable by bioluminescence imaging (BLI), CAR-T cells ($10 \times 10^6$ cells in 100 μL PBS) were injected i.v. into the tail veins of the NSG mice. Dimerization of the ON-switch AvidCAR was induced by intraperitoneal (i.p.) administration of 2 mg/kg AP20187 (Takara). AP20187 was prepared by dissolution in ethanol yielding a concentration of 12.5 mg/mL and further dilution to the final working concentration of 0.5 mg/mL in vehicle solution [4% ethanol, 10% PEG-400 and 1.7% Tween-80 (both Sigma-Aldrich) in water]. Those working stocks of AP20187 were sterile filtered and used for injection within 30 min. Injection of only the vehicle solution served as control. The mice were monitored daily and sacrificed by cervical dislocation at the first sign of disease (weight loss, rough fur, beginning paralysis).

**In vivo bioluminescence imaging.** BLI was performed at the PIL (Medical University of Vienna) using an IVIS Spectrum In Vivo Imaging System (Perkin Elmer). D-luciferin (Perkin Elmer) was freshly dissolved in PBS to a final concentration of 15 mg/mL and sterile filtered. Mice were anesthetized with isoflurane (Abbvie) and received i.p. injections of the luciferin solution (dosage of 150 mg/kg body weight). After 15–20 min, mice were transferred to the IVIS Imaging System and bioluminescence was measured immediately in medium binning mode with an automatic acquisition time (ranging from 1 s to 2 min) to obtain unsaturated images. The total photon flux was determined within the region of interest which encompassed the entire body of the mouse. Tumor growth was monitored every 3–4 days. Living Image Software (Caliper) was used to analyse the data.

**In vivo model for the AND-gate AvidCAR.** NSG mice were injected via tail vein with $0.5 \times 10^6$ cells of a 1:1:1:1 mixture of Nalm-6, hEGFRt-Nalm-6, hHER2t-Nalm-6 and hHER2t-hEGFRt-Nalm-6 cells and 3 days later with $10 \times 10^6$ CAR-T cells or mock-T cells, as indicated. Mice were sacrificed 13 days after tumor injection. Bone marrow single-cell suspensions were obtained by flushing one femur with ~10 mL sterile PBS and passing through a 70 μm cell strainer (Corning Falcon). After disruption of the liver through a 70 μm cell strainer and several washing steps, lymphocytes were separated from hepatocytes using a 33.75% Percoll gradient (GE Healthcare). Spleen single-cell suspensions were obtained by two consecutive passages through 70 μm cell strainers. Blood was drawn retro-orbitally from isoflurane-anesthetized mice and collected in EDTA-containing tubes (Greiner). Fifty μL blood was used for the staining with fluorescent antibodies, followed by red blood cell lysis using an RBC lysis/fixation solution (Biolegend).

**Mathematical modeling.** The multivalent interaction between a bivalent antigen and a bivalent CAR that can undergo kinetic proofreading leads to a large number of distinct chemical states. Therefore, it is not practical to manually enumerate them. To overcome this, we used the rule-based framework of BioNetGen[70] to define five elementary reaction: intermolecular binding ($k_{on}$), intermolecular unbinding ($k_{off}$), intramolecular binding ($\sigma \, k_{on}$), and the kinetic proofreading phosphorylation rate ($k_p$). We have assumed that (1) the intramolecular unbinding rate (i.e., the unbinding rate between a single CAR and antigen when both CARs are bound to antigen) is equal to the intermolecular unbinding (i.e., unbinding rate of a single CAR and antigen when only a single CAR in the dimer is bound to antigen) and (2) the kinetic proofreading phosphorylation rate can only take place when at least one CAR in the dimer is bound to antigen. When a single CAR in the dimer is bound, the binding rate of the second CAR in the dimer to the second antigen in the dimer is expected to be dependent on the intermolecular on-rate (since the same binding interface is formed) multiplied by a factor that accounts for the local concentration (and potentially other factors, such as structural alignment) that we define to be σ with units of concentration. This parameter controls the avidity of the CAR with $\sigma = 0$ producing a monovalent CAR and increasing values of σ lead to increasing avidity. Collectively, these elementary reactions lead BioNetGen to 72 chemical states that were exported as a system of 72 coupled nonlinear coupled ordinary differential equations. The initial conditions were 0 for all chemical species except the concentration of the CAR ($1,000/\mu m^2$) and antigen (varied as indicated). The parameter values used were $k_{on} = 5 \times 10^{-4} \, \mu m/s$, $k_{off} = 10/s$, and $k_p = 0.1/s$. These equations were solved numerically in Matlab (Mathworks, MA) to produce the panels in Supplementary Fig. 3b showing the bound receptor and phosphorylated receptor. To calculate the cellular response in Supplementary Fig. 3b, we have assumed a saturating Hill function connecting phosphorylated receptor to the cellular response using a threshold of ($k_{thres} = 2/\mu m^2$). Although the parameter values are expected to alter the absolute concentration of antigen inducing receptor binding, phosphorylation, and cellular response, they do not alter our main qualitative conclusion which is that increasing the cooperativity factor of dimeric binding (σ) increases the amount of phosphorylated receptor and the cellular response without increasing CAR binding. The bngl file defining the model along with the m-file used to solve it can be found in Supplementary Data 4 that also includes all parameter values.

**Statistical analysis.** Statistical analysis was performed using GraphPad Prism 7 software for Windows (GraphPad Software Inc.) and SAS software (Version 9.4, SAS Institute). Graphs were generated using GraphPad Prism 7 software. Data are presented as individual data points or as means ± SD. Statistical analysis was performed using two-tailed paired $t$-tests, two-tailed ratio-paired $t$-test, or two-tailed unpaired $t$-test, as indicated. Statistical analysis of Figs. 4b and 6b was done with a mixed model to compare fixed-effect means taking into account that the data are correlated. Statistical analysis of Figs. 5d and 6d was done by log-rank analysis or two-way ANOVA, respectively.

**Reporting summary.** Further information on research design is available in the Nature Research Reporting Summary linked to this article.

## Data availability

The authors declare that all data supporting the findings of this study are available within the article and its Supplementary information files or from the corresponding authors upon reasonable request. The source data for the Figs. 2a, c, e, 3b, e, g, 4b, d, 5a–c, and 6b, d, as well as Supplementary Figs. 2c, 4a, b, d, 5e, 6a, b, and 12c, e are provided as a Source data file. Source data are provided with this paper.

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

## Acknowledgements

We would like to acknowledge Ulrike Pötschger for assistance in statistical analysis. The SPR equipment was kindly provided by the EQ-BOKU VIBT GmbH and the BOKU Core Facility for Biomolecular & Cellular Analysis. We are thankful to the excellent support of the Preclinical Imaging Laboratory (PIL) and the Center for Biomedical Research of the Medical University of Vienna. This work is supported by the Austrian Science Fund (FWF Project P32001-B and FWF Project W1224—Doctoral Program on Biomolecular Technology of Proteins—BioToP), the Federal Ministry for Digital and Economic Affairs of Austria, and the National Foundation for Research, Technology and Development of Austria to the Christian Doppler Research Association (Christian Doppler Laboratory for Next Generation CAR T Cells) and by private donations to the Children's Cancer Research Institute (Vienna, Austria). O.D. is supported by a Wellcome Trust Senior Research Fellowship (207537/Z/17/Z). T.P. and J.B.H. were supported by the European Union's Horizon 2020 research and innovation programme under the Marie Skłodowska-Curie grant agreement no. 721358.

## Author contributions

B.S., C.M.S., C.U.Z., T.P., and M.A.S. performed the experiments and analyzed data. B.S., E.M.P., J.B.H., M.W.T, and M.L. designed the experiments and critically revised the data. B.K., M.C.B., and E.M.P. assisted in animal models. E.L. contributed valuable reagents and helped with experimental design. O.D. assisted in mathematical modeling. B.S., O.D., C.O., W.H., M.W.T., and M.L. wrote the paper. All authors edited and approved the paper.

## Competing interests

B.S., C.U.Z., C.O., M.W.T., and M.L. have filed three patent applications that relate to the technology described in this paper: patent applicant: St. Anna Kinderkrebsforschung (Zimmermannplatz 10, 1090 Vienna, Austria), Universität für Bodenkultur Wien (Gregor Mendel-Strasse 33, 1180 Vienna, Austria); name of inventor(s): B.S., M.W.T., M.L.; application number: PCT/EP2019/076917 (WO 2020/070290 A1); status of application: pending; specific aspect of paper covered in patent application: AvidCARs with inducible function. Patent applicant: St. Anna Kinderkrebsforschung (Zimmermannplatz 10, 1090 Vienna, Austria); Universität für Bodenkultur Wien (Gregor Mendel-Strasse 33, 1180 Vienna, Austria); name of inventor(s): B.S., M.W.T., M.L.;. application number: PCT/EP2019/076916 (WO 2020/070289 A1); status of application: pending; specific aspect of paper covered in patent application: AvidCARs for combinatorial antigen recognition (AND-gate function). Patent applicant: St. Anna Kinderkrebsforschung (Zimmermannplatz 10, 1090 Vienna, Austria), Universität für Bodenkultur Wien (Gregor Mendel-Strasse 33, 1180 Vienna, Austria); name of inventor(s): Charlotte Brey*, M.T., C.O., M.L.; application number: PCT/EP2018/086299 (WO 2019/122188 A1); status of application: pending; specific aspect of paper covered in patent application: alternative ON-switch system for conditional heterodimerization of AvidCARs (mentioned in the "Discussion") (*now C.U.Z.). The remaining authors declare no competing interests.
