## [Peer Review File · Nature Communications]

Reviewers' Comments:

Reviewer #1:

Remarks to the Author:

Salzer et al. present a new spin on the CAR platform that is based on coupling reduced-affinity targeting domains to exogenous CAR dimerization. The authors propose this as a better system than other published modalities for combinatorial targeting.

Overall, the engineering work is nicely done and well thought-out. However:

Major comments:

1. The concept needs to be fleshed out with more examples of targeting domains. I do not think it is sufficient to use only the monovalent E11.4.1 binder. Also, more models with scFv to show how generalizable this concept is.
2. One of the major claims here is that the AVID-CAR system can discriminate single- from double-antigen-expressing cells. This is done quite well in vitro but the in vivo work is not convincing. The authors would need to show in the same animal that single antigen tumor cells are ignored while double antigen cells are killed. The claim on page 12 line 282-283 is not actually supported by their data.
3. Page 5, lines 112-124. This claim would be greatly enhanced by actual experimental data to test their mathematical prediction.
4. Page 6, line 136. Since the data suggest that the low affinity AvidCAR confers high sensitivity only in settings where the ligands are present in dimeric form, this makes me question the relevance of this platform to real life. What two different tumor antigens are present in dimeric form?
5. Page 11 lines 262-264 and Fig 5d,e. Here the results are quite marginal and the follow-up is short. Therefore the data are not particularly convincing.
6. Page 12, line 301-302. There actually are examples of bispecific CARs that show "AND" gating logic. For example, Kloss et al, Nat Biotech 2013.
7. Page 13, line 320-322. Increasing tumor specificity by requiring a dependence on two different antigens (i.e. AND gating) opens not one but two doors for tumor escape through antigen loss (or downmodulation). Therefore I am not sure that this design is ultimately advisable.

Minor comments:

1. It would be interesting to see more in-depth mechanistic studies. For example, page 8 line 185, are there differences in the activation of signaling proteins that might explain why the different designs have disparate effects on killing vs cytokine production?
2. Fig 2d. although there is no statistical significance in the high and intermediate affinity comparisons, I wonder whether with additional repeats of this experiment, these results will hold up. There is a clear trend to reduced activity in the ser-BBz constructs.

Reviewer #2:

Remarks to the Author:

One of the major challenges of CART therapy is controlling the toxicities associated with the therapy due to the lack of cancer specific targets and treatment associated side effects, such as tumor lysis syndrome and cytokine storm, as well as off target effects. Developing novel ON-switch and AND-gate systems that can not only effectively control specificity of CARs, but also minimally affect efficacy of the therapy is critically important to advance the field of CART therapy. In this report, the authors developed a novel avidity-controlled CAR (AvidCAR) platform with inducible and logic control functions that enable to control CART function via ON-switch and AND-gate mechanisms.

However, the data present in current version of manuscript are far from being convincing and solid

enough to show that the systems are optimized to be potentially useful. Major changes have to be made before publication.

Some suggestion for revision:

1. Based on the data shown in Figure 2F and Supp. 4B, that suggest: a. Overall, AvidCAR activities are lower than high-affinity CAR; b. At lower antigen density, AvidCAR still reactive. C. At high antigen density, there is nearly no differences between AvidCAR and monomeric low affinity Ser-CAR for both lytic and cytokine production. Therefore, it seems that monomeric low affinity Ser-CAR is safer and more specific to high antigen density (equivalent to tumors), where spares low antigen density (equivalent to normal tissues).

2. The tumor lines that express EGFR or Her2/neu at different levels should be tested to demonstrate that AvidCAR can differentially recognize tumors that express antigen at high levels.

3. The authors conclude that AvidCAR cannot use scFv, but only in Fv form, which might be misleading, since only one scFv (4D5-5) was tested.

4. AvidCAR works with only 4-1bb domain based CAR, which also imply that the systems can be affected by some unpredicted and uncontrollable factors and makes this platform less reliable and useful.

5. Only Her2/neu high expressing cell lines were used in Affibody mutant R10A AvidCAR work. Tumors, or cell lines that express Her2/neu at different levels should be tested, to exclude the possibility that the dimerized R10A AvidCAR has no discrimination for antigen expressions.

6. Although cytokine productions were used in some experiments, the in vitro test was mainly relayed on CTL (killing) assay. All the CTL assays were used fixed single E:T ratio, which cannot exactly tell the difference of the killing ability of T cells, especially at high E:T ratio, such as for Figure 5A&B, E:T ratio of 10:1 was used. The lytic activity at different E:T ratio should be shown for most of the experiments.

7. The Nalm6 model used in this report was much weaker than most of the reports by other people who used 1e6 Nalm6 (i.v.) and treatment started at day 5 or 7 later with 1 to 10 million of transduced CART cells. While in this report, 5e5 Nalm6 with treatment started at 3 days (Figure 5C) or 5 days (Figure 6C) with 10 million of CART cells, with such condition, 10 million FMC63 CD19 CART cells are far more than enough to completely cure the mice, while AvidCAR only showed minimum effect on controlling tumor growth. That's might be the reasons why the experiments (for both Figure 5C and 6C) were terminated at extremely early after treatments.

8. For Nalm6 model, long time (usually two to three month) follow-up and survival cures are required.

9. Since the systems are mainly developed for the treatment of solid tumors and the targets developed in this report are EGFR and Her2/neu, the system should be tested in solid tumor models.

Responses to the reviewers' comments

We thank the reviewers for their careful reading of the manuscript and their helpful comments. Based on the comments, we have improved and clarified the manuscript and also added several *in vitro* and *in vivo* experiments. Please find below a point-by-point response to all comments. The corresponding changes in the text of the revised manuscript (including figures) are marked in yellow (uploaded in the Supplementary Information as "Article (Salzer et al.) marked changes" and "Figures (Salzer et al.) marked changes").

Reviewers' comments:

Reviewer #1:

Salzer et al. present a new spin on the CAR platform that is based on coupling reduced-affinity targeting domains to exogenous CAR dimerization. The authors propose this as a better system than other published modalities for combinatorial targeting.

Overall, the engineering work is nicely done and well thought-out. However:

Major comments:

1. The concept needs to be fleshed out with more examples of targeting domains. I do not think it is sufficient to use only the monovalent E11.4.1 binder. Also, more models with scFv to show how generalizable this concept is.

Obviously, we did not sufficiently highlight our second targeting domain (an affibody-based HER2-specific binder). This additional targeting domain was successfully used both in homodimeric AvidCARs (Suppl. Fig. 6a) and in heterodimeric AND-gate AvidCARs together with the EGFR-specific rcSso7d-based binder (Figs. 4a, 4b and Figs. 6a-d). Nevertheless, in the course of the revision process, we have additionally engineered two new AvidCARs based on nanobodies with specificity against GFP and HER2, respectively. Suppl. Fig. 12 shows that the function of these new nanobody-AvidCARs strongly depends on bivalent interaction and can be controlled by conditional homodimerization by AP20187, thus confirming our data generated with the rcSso7d-based binder against EGFR and the HER2-targeting affibody. In total, we have successfully constructed AvidCARs with four different targeting domains based on three different protein scaffolds. We mention this in the revised manuscript in the Discussion (lines 327-332) and have added the new data as new Suppl. Fig. 12.

Regarding the scFvs, we would like to note that we did *not* want to present the separation of VH and VL onto two separate CAR molecules (Fig. 3d) as a solution to the use of scFvs that tend to cluster. We do not believe that this split Fv expression format is useful for real world applications. Instead, this was only an *in vitro* model system to study scFv clustering and its effect on AvidCAR function. Moreover, we would like to note that we do not claim that all scFv variants cluster on the surface of CAR T cells. For example, in lines 183-184, we write: "Together, these data demonstrate that intermolecular VH/VL-dimerization of at least certain scFvs can cause CAR oligomerization." To prevent any misunderstandings, we extended the discussion on scFv clustering in lines 332-351.

2. One of the major claims here is that the AVID-CAR system can discriminate single- from double-antigen-expressing cells. This is done quite well *in vitro* but the *in vivo* work is not convincing. The authors would need to show in the same animal that single antigen tumor cells are ignored while double antigen cells are killed. The claim on page 12 line 282-283 is not actually supported by their data.

We agree that in the original manuscript the AND-gate AvidCAR approach was not sufficiently supported by *in vivo* experiments. In particular, the reviewer referred to the notion that our AvidCAR system can distinguish between single- and double-positive target cells without the need for “safe distances” (p.12 line 282-283 in the original manuscript). We therefore designed an additional *in vivo* experiment, where we mixed four different Nalm-6 populations (hEGFRt/hHER2t double-positive Nalm-6, both types of single-positive controls and double-negative versions) 1:1:1:1 and administered this mix of Nalm-6 cells intravenously to NSG mice. Thus, the specificity for double-positive target cells vs. single positive or double negative controls could be analyzed within the same animal. In addition, we compared our AND-gate AvidCAR with a reference CAR containing the high-affinity version of the α EGFR binder (positive control) and T cells expressing no CAR (negative control). On day 13, i.e. 10 days after T cell inoculation, we determined the ratios and absolute cell counts of the four Nalm-6 populations in blood, bone marrow, spleen and liver.

Figs. 6c, 6d and Suppl. Fig. 10 show very convincingly that AvidCAR-T cells specifically killed the double-positive Nalm-6 cells while ignoring the single-positive Nalm-6 cells, even though they were present in the same tissue of the same animal. Please note that this high specificity was obtained despite even higher antigen expression levels in the single positive controls (Suppl. Fig. 8d). In addition to the high specificity, the efficacy of T cells expressing the AND-gate AvidCAR was comparable to T cells expressing the high-affinity α EGFR-CAR (equally efficient in the liver; only slightly less efficient in bone marrow). No Nalm-6 cells were detectable in blood or spleen.

In summary, the new experiment with the mixed Nalm-6 populations yielded several important findings:

- 1) The AND-gate AvidCAR is highly specific for double-positive target cells and does not require “safe distances” (no significant killing of single-positive Nalm-6 cells in bone marrow and liver). Notably, this result was obtained with single-positive Nalm-6 cells that expressed the antigen at even higher levels than the double-positive Nalm6 cells.
- 2) The AND-gate AvidCAR is efficient (same efficacy as the high-affinity α EGFR-CAR in liver, slightly lower efficacy than the reference CAR in bone marrow).
- 3) The hEGFR/hHER2-Nalm6 clone is significantly more aggressive. This clone was enriched in the liver 13 days after inoculation approximately 20-50 times compared to single-positive Nalm-6 cells and was also incompletely eliminated even by the high-affinity α EGFR-CAR-T cells (AvidCAR-T cells had the same efficacy). This explains in retrospect why AvidCAR-T cells only moderately delayed the outgrowth of this Nalm-6 clone in the original experiment, where killing of different target cells was analyzed in different animals (Fig. 6c of the original manuscript; BLI-monitoring).

In addition to the new Figs. 6c, 6d, and Suppl. Figs. 9-11, also the text in lines 294-310 was adapted.

3. Page 5, lines 112-124. This claim would be greatly enhanced by actual experimental data to test their mathematical prediction.

For immune receptors the influence of the lifetime of interaction on phosphorylation events and signal transduction is well established [Goldstein et al., Nat. Rev. Immunol. (2004), PMID: 15173833]. The lifetime of an interaction is determined by K_{off} , i.e. the rate constant for the dissociation of a protein from its ligand. The dissociation of a bivalent protein occurs in two steps and the epitope released first can bind again. Therefore, the dissociation is much slower than that of a monovalent protein. The effect of the synergy of multiple interactions (avidity-effect) is the basis for our concept of integrating ON-switch and AND-gate function into CARs. With our mathematical model we can simulate the influence of the avidity-effect on T cell activation and e.g. show that CARs with higher avidity can activate T cells at lower antigen densities (Suppl. Fig. 3b, diagram no. 3).

It might not have been highlighted sufficiently, that our experimental data in Fig. 2f do confirm this correlation. First, the switchable AvidCAR in its monovalent off-state (no avidity effect, $\sigma = 0$) is

not activated efficiently, even at high antigen densities. Second, antigen sensitivity of the dimeric (switched on) AvidCAR is considerably improved upon additional dimerization of the antigen (i.e., by enhancing the avidity effect), thus reflecting the data from the mathematical model. For better comprehensibility we have made textual corrections in the manuscript (lines 123-128 and 145-147).

4. Page 6, line 13. Since the data suggest that the low affinity AvidCAR confers high sensitivity only in settings where the ligands are present in dimeric form, this makes me question the relevance of this platform to real life. What two different tumor antigens are present in dimeric form?

We do not consider high antigen sensitivity as a mandatory prerequisite for the applicability of our AND-gate approach. Instead, we also consider the targeting of highly-expressed monomeric antigens to be attractive. In fact, especially among the most advanced CAR therapies there are target antigens that are highly overexpressed, but are also rather unspecific for the tumor (e.g. HER2, MUC1, EGFR, CD56, CD138, GD2, EPCAM) [MacKay et al., Nat. Biot. (2020), PMID: 31907405]. In order to spare tissues expressing such antigens at lower levels, the field considers “detuning” of the CARs, i.e., the reduction of antigen sensitivity by decreasing the binding affinity [Watanabe et al. (2018), PMID: 30416506; Arcangeli et al. (2017), PMID: 28479045; Du et al. (2017), PMID: 30753824; Majzner et al. (2019), PMID: 30655315; Caruso et al. (2015), PMID: 26330164; Liu et al. (2015), PMID: 26330166]. Our AND-gate AvidCAR also needs a high antigen density (if the antigens are not colocalized). However, this CAR is not activated if only one of the two target antigens is expressed, even if the density of this antigen is higher than on the double-positive targets (see new *in vivo* experiment in Figs. 6c, 6d, Suppl. Fig. 8d and Suppl. Fig. 10). This is a crucial advantage over the conventional affinity-tuning approach, which could not generate such high specificity.

Nevertheless, we are aware of the risk of antigen downregulation on tumor cells and the advantage of high antigen-sensitivity. Therefore, co-localized antigens are of course particularly attractive candidates. We added a more comprehensive discussion on co-localized antigens in lines 400-403.

5. Page 11 lines 262-264 and Fig 5d,e. Here the results are quite marginal and the follow-up is short. Therefore the data are not particularly convincing.

Unfortunately, during the revision process, we had to cancel a planned repetition of the experiment with a longer follow-up due to the shut-down on our campus because of COVID-19. As we agree with the reviewer that the data in the original Figures 5d-e were only secondary and not thoroughly elaborated, we propose to show the data in the supplemental data and to tone down the statement in the revised manuscript (Suppl. Fig. 8a and line 274). However, we repeated the original experiment shown in Figure 5d and compared the long-term survival of mice after AvidCAR-T cell treatment in the continuous presence of dimerizer (“on”-state) in order to address a point raised by the other reviewer. These new data (now included as new Fig. 5d) confirm that the AvidCAR has no measurable background activity in its “off”-state (compared to mock T cells without CAR), while it significantly prolongs survival in its “on”-state (i.e., in the presence of the dimerizer) (new Fig. 5d and lines 276-280).

6. Page 12, line 301-302. There actually are examples of bispecific CARs that show “AND” gating logic. For example, Kloss et al, Nat Biotech 2013.

Obviously, we did not explain appropriately that we were referring to single bispecific CAR molecules which can only trigger T cells upon simultaneous binding of two antigens. We changed the text in the revised manuscript accordingly (lines 361-362). The approach presented by Kloss et al. is not based

on a single bispecific CAR but on two separate receptors, i.e., a 1st generation CAR against antigen A and a chimeric co-stimulatory receptor against antigen B. Such AND-gates based on separate transduction of signals 1 and 2 by antigens A and B, respectively, may be less specific, since, for example, substantial cytotoxicity is triggered by signal 1 only, resembling a 1st generation CAR.

7. Page 13, line 320-322. Increasing tumor specificity by requiring a dependence on two different antigens (i.e. AND gating) opens not one but two doors for tumor escape through antigen loss (or downmodulation). Therefore I am not sure that this design is ultimately advisable.

We agree that antigen loss is a major problem in CAR T cell therapy in general and probably even more so with any AND-gate approach that targets two tumor antigens. However, analogous to OR-gate approaches with conventional CARs, there is the possibility to concomitantly target two or more alternative combinations of antigens, which would strongly reduce the risk of antigen escape.

Moreover, it has been shown in mouse models that efficient CAR T cell responses can lead to epitope spreading, i.e., endogenous T cell responses against other tumor antigens. In the clinical setting, the activation of such endogenous immune responses is likely to be prevented by lymphodepleting conditioning prior to the administration of CAR T cells. However, it is conceivable that future strategies might enable the application of CAR T cells without prior lymphodepletion, which could trigger epitope spreading and thereby impede immune escape through antigen loss. Of note, improving tumor specificity (e.g. by using AND-gate approaches) will allow for implementation of more potent CAR T cell strategies, which might further enhance the endogenous immune response in the tumor.

Minor comments:

1. It would be interesting to see more in-depth mechanistic studies. For example, page 8 line 185, are there differences in the activation of signaling proteins that might explain why the different designs have disparate effects on killing vs cytokine production?

The observed differences are interesting for us as well. However, a mechanistic elucidation is outside the scope of the proof-of-concept study presented here.

2. Fig 2d. although there is no statistical significance in the high and intermediate affinity comparisons, I wonder whether with additional repeats of this experiment, these results will hold up. There is a clear trend to reduced activity in the ser-BBz constructs.

We agree with the reviewer. Indeed, a reduced activity of monomeric serine-CARs (ser-BBz) compared to dimeric cysteine-CARs is to be expected, since the avidity-effect also occurs at higher affinities of the binding domains. Nevertheless, the lower the affinity, the stronger the dependency on multivalent interaction (i.e., avidity) will be, which is reflected in the data shown in Fig. 2d. The most important point was to find an affinity range, where the dependency on dimerization is as pronounced as possible and where the activity of the monomeric CAR is reduced to background levels. Even though the high- and intermediate-affinity CARs might be dependent on dimerization, they still show considerable activity in their monomeric versions, potentially causing leaky AvidCAR function. Therefore, we chose the low affinity version for further experiments.

Reviewer #2 (Remarks to the Author):

The one of major challenges of CART therapy is controlling the toxicities associated with the therapy due the lack of cancer specific targets and treatment associated side effects, such as tumor lysis syndrome and cytokine storm, as well as off target effects. Developing novel ON-switch and AND-gate systems that can not only effectively control specificity of CARs, but also minimally affect efficacy of the therapy is critically important to advance the field of CART therapy.

In this report, the authors developed a novel avidity-controlled CAR (AvidCAR) platform with inducible and logic control functions that enable to control CART function via ON-switch and AND-gate mechanisms.

However, the data present in current version of manuscript are far from being convincing and solid enough to show that the systems are optimized to be potentially useful. Major changes have to be made before publication.

Some suggestion for revision:

1. Based on the data shown in Figure 2F and Supp. 4B, that suggest: a. Overall, AvidCAR activities are lower than high-affinity CAR; b. At lower antigen density, AvidCAR still reactive. C. At high antigen density, there is nearly no differences between AvidCAR and monomeric low affinity Ser-CAR for both lytic and cytokine production. Therefore, it seems that monomeric low affinity Ser-CAR is safer and more specific to high antigen density (equivalent to tumors), where spares low antigen density (equivalent to normal tissues).

It seems that the legend of the crucial Fig. 2f unfortunately was insufficient or misleading and therefore the lines might have erroneously been assigned to the wrong conditions (e.g., black solid line wrongly assigned to the "monomeric low affinity Ser-CAR"?). We apologize for this obviously misleading figure legend. We fear that this misunderstanding was also the reason for the criticism in points 2 and 5 and significantly contributed to the fact that our concept as a whole did not appear convincing.

Our concept is based on the fact that AvidCARs can activate T cells efficiently only upon bivalent, but not monovalent interaction. An ON-switch AvidCAR should therefore only be functional after its dimerization by the respective dimerizer. This is exactly what Fig. 2f shows (right diagram). The AvidCAR in the "off"-state, i.e. the monomeric low affinity Ser-CAR (*dotted* lines; *black and red*) has hardly any activity even at the highest antigen density. Only the AvidCAR in the "on"-state, i.e. homodimerized (*solid* lines; *black and red*), can efficiently activate T cells. While the dimeric AvidCAR ("on"-state) is also reactive against monomeric antigen (black solid line), its antigen sensitivity is further increased upon antigen dimerization (red solid line). That is, in Fig. 2f the lines of *one* color (solid versus dotted lines) must be compared to each other in order to compare the function of the AvidCAR in the "on"- and "off"-state. We apologize for the confusion. We modified the legend of Fig. 2f and the text in the results (lines 137-144) in the revised manuscript to prevent any confusion of the readers.

We would also like to note that Fig. 2f demonstrates how the avidity-effect can be used to generate an ON-switch function. Our primary goal was *not* to improve specificity by sensing antigen density. The ability for sensing antigen density results from the dependence on bivalent interaction when using low-affinity binding domains and is only a by-product.

To improve *specificity*, we instead used the avidity-effect for *combinatorial recognition* of antigens A and B (Figs. 4a, 4b and Fig. 6). Just as the ON-switch AvidCAR in Fig. 2f was only active as a dimer, the dependence on bivalent interaction in a *bispecific* AvidCAR leads to robust AND-gate function. The new Figs. 6c, 6d and Suppl. Fig. 10 now show very convincingly *in vivo*, that a bispecific AvidCAR

exclusively targets double-positive target cells while ignoring single-positive cells. In this new *in vivo* experiment, the single-positive cells were spared, although these cells expressed the antigen at even higher levels than the double-positive cells (new Figs. 6c, 6d, Suppl. Figs. 8d and Suppl. Fig. 10, and lines 294-310 in the result section). This would not have been possible with an approach purely based on affinity-tuning.

2. The tumor lines that express EGFR or Her2/neu at different levels should be tested to demonstrate that AvidCAR can differentially recognize tumors that express antigen at high levels.

We assume that also this point is based on the conclusion that “at high antigen density there is nearly no differences between AvidCAR and monomeric low affinity Ser-CAR” (as stated in point 1C). We can only explain this conclusion from Fig. 2f by an erroneous assignment of the monomeric AvidCAR to the black solid line, which was then compared to the red solid line.

Fig. 2f in fact shows the dependence of AvidCAR activation on antigen density, whereby for a dimeric target antigen a lower density is required than for a monomeric antigen (red solid line vs. black solid line). However, as stated above, the main point is the dependency of the AvidCAR on bivalent antigen recognition, i.e., on the avidity-effect.

3. The authors conclude that AvidCAR cannot use scFv, but only in Fv form, which might be misleading, since only one scFv (4D5-5) was tested.

Also this point shows us that our original manuscript was not clear enough at some important parts. In fact, we did not want to present the separation of VH and VL onto two separate CAR molecules (Fig. 3d) as a solution to the use of scFvs that tend to cluster. We do not believe that this Fv expression format is useful for real world applications. Instead, this was only an *in vitro* model system to study scFv clustering and its effect on AvidCAR function. Also, we did not want to claim that all scFvs oligomerize on the surface of CAR T cells (and are therefore not suited). We explained this in lines 332-351 and additionally rephrased lines 324-326 of the discussion: “(iii) the use of single-domain antigen-binding domains, which seems to be preferable to exclude potential oligomerization that has been observed for some scFvs”. Please see also lines 340-341: “However, we do not exclude that other scFvs may be suitable for the construction of AvidCARs.”

4. AvidCAR works with only 4-1bb domain based CAR, which also imply that the systems can be affected by some unpredicted and uncontrollable factors and makes this platform less reliable and useful.

This statement applies to the CAR field in general. For example, tonic signaling with many scFv-based CARs is a factor that was originally not foreseen and is still difficult to control. There are many more examples, and they all underline the need for continuous improvement of CAR-T cell therapy at all levels.

In our opinion, the influence of signaling domains does not make the AvidCAR platform less reliable or useful. However, it is important to investigate which signaling domains are suitable for use in AvidCARs. The optimization of signal transduction in CARs is being intensively pursued in the field and seems particularly relevant in our context.

We have revised Fig. 3g and Suppl. Fig. 5e and now show that the signaling domain of CD2 is also very well suited for use in AvidCARs. For confidentiality reasons, we could not show this data at the time of initial submission. We also adapted the text in lines 191, 194-196 and 198-199.

5. Only Her2/neu high expressing cell lines were used in Affibody mutant R10A AvidCAR work. Tumors, or cell lines that express Her2/neu at different levels should be tested, to exclude the possibility that the dimerized R10A AvidCAR has no discrimination for antigen expressions.

Like point 2, this comment seems to be based on the erroneous conclusion that “at high antigen density there is nearly no differences between AvidCAR and monomeric low affinity Ser-CAR” (as stated in comment 1C). We would therefore like to refer to our answers to points 2 and 1.

Again, it is crucial for our concept that the HER2-specific AvidCAR, like the EGFR-specific AvidCAR, remains inactive during monovalent interaction even at high antigen density and only triggers during bivalent interaction (which is shown in Suppl. Fig. 6a).

6. Although cytokine productions were used in some experiments, the in vitro test was mainly relayed on CTL (killing) assay. All the CTL assays were used fixed single E:T ratio, which cannot exactly tell the difference of the killing ability of T cells, especially at high E:T ratio, such as for Figure 5A&B, E:T ratio of 10:1 was used. The lytic activity at different E:T ratio should be shown for most of the experiments.

We agree and are aware that different E:T ratios can have an impact on the experiment. Therefore, the experiment mentioned by the reviewer (Figs. 5a and 5b) was originally performed with two different E:T ratios (10:1 and 2:1), whereby the effects observed with those two E:T ratios were highly comparable (apart from the generally elevated killing levels in the 10:1 condition, as expected). The revised manuscript now also shows the 2:1 condition (Figs. 5a, 5b and line 269).

7. The Nalm6 model used in this report was much weaker than most of the reports by other people who used 1e6 Nalm6 (i.v.) and treatment started at day 5 or 7 later with 1 to 10 million of transduced CART cells. While in this report, 5e5 Nalm6 with treatment started at 3 days (Figure 5C) or 5 days (Figure 6C) with 10 million of CART cells, with such condition, 10 million FMC63 CD19 CART cells are far more than enough to completely cure the mice, while AvidCAR only showed minimum effect on controlling tumor growth. That's might be the reasons why the experiments (for both Figure 5C and 6C) were terminated at extremely early after treatments.

We repeated the experiment shown in Fig. 5c and monitored survival over a prolonged time span. Also the FMC63 CD19-CAR-T cells did not cure the mice in our hands, as can now be seen in the new *in vivo* experiment (Fig. 5d; please see comment to point 8 for a detailed description of this experiment). Importantly, the Avid-CAR in the “on”-state was comparably effective as the control CARs (FMC63 α CD19 and high-affinity α EGFR). Similarly, the AND-gate AvidCAR showed comparable efficacy to the high-affinity control CAR (Figs. 6c and 6d) in the other *in vivo* experiment that was added in the course of this revision process. Together, those two newly added *in vivo* experiments (Fig. 5d and Fig. 6c and d) both demonstrate that activated AvidCARs (being dimeric and interacting via both antigen targeting domains) are almost as efficient as high-affinity control CARs *in vivo*.

We attribute the decreased efficacy of the CD19-CAR-T cells (as well as the efficacies other CAR-T cells studied here) to the expansion conditions of our CAR-T cells. In our protocol, the α CD3/CD28 beads are not removed and the T cells are only administered 15-17 days after stimulation when the expansion already flattens. We assume that *in vivo* our CAR-T cells expanded at a correspondingly reduced rate. This connection is known from other studies [Kagoya et al. (2017), PMID: 28138559; Ghassemi et al. (2018), PMID: 30030295]. We will improve our T cell expansion protocol for our future work. Nevertheless, since all CAR T cells in this study were expanded and generated using the same protocol, their efficacies can be compared to each other, showing that – as an example – activated AvidCARs are almost as potent as their high-affinity controls.

Regarding the seemingly limited efficacy of the AND-gate CAR in the original *in vivo* experiment (old Fig. 6c), we would like to refer to the new *in vivo* experiment shown in Figs. 6c, 6d and Suppl. Fig. 10, and described in lines 294-310. This new experiment shows that the EGFR/HER2-transduced Nalm-6 target cell clone is substantially more aggressive than the double-negative wild-type and single-positive Nalm-6 controls, as shown in the new Figs. 6c and 6d. 13 days after inoculation, the EGFR/HER2-Nalm-6 clone was enriched in the liver approximately 50 times compared to HER2-Nalm-6 cells and 22 times compared to EGFR-Nalm-6 cells in the mock T cell group. Especially in the liver it is evident that even the high-affinity EGFR-CAR-T cells, which were comparably effective as the AND-gate AvidCAR-T cells, could not completely eliminate the cells of the EGFR/HER2-Nalm-6 clone. In retrospect, this probably explains the small therapeutic effect in the old Fig. 6c, for which the same target cell clone was used.

8. For Nalm6 model, long time (usually two to three month) follow-up and survival cures are required.

We have repeated the experiment with the dimerizer-dependent α hEGFR-AvidCAR (old Fig. 5c) and now show the survival curves in the new Fig. 5d. As already mentioned in point 7 above, the control CARs also exhibited a relatively moderate efficacy, resulting in death of all mice by the end of week 6. The efficacy of the α hEGFR-AvidCAR in the “on”-state was comparable to the control-CARs as in the original experiment (Fig. 5c). Importantly, in the “off”-state the α hEGFR-AvidCAR had no activity, as demonstrated by the comparison with mock T cells without CAR. This could not be deduced in the original experiment, where PBS was used as a negative control (Fig. 5c). The new experiment is described in lines 276-280. The original Fig. 5d is now Suppl. Fig. 8a.

9. Since the systems are mainly developed for the treatment of solid tumors and the targets developed in this report are EGFR and Her2/neu, the system should be tested in solid tumor models.

To address this point, we had planned another *in vivo* experiment in the course of this revision process. But because of the shutdown of our facility due to COVID-19, we unfortunately could not establish a solid tumor model in our lab. In addition, the mouse breeding on the campus had to be shut down and even in the best-case scenario, we could have additional *in vivo* data not before September.

We would like to note that we did conduct a preliminary *in vivo* experiment in order to establish the solid tumor model. However, in that initial experiment we did not see a clear efficacy even with the high-affinity control CAR-T cells (see Fig. A below at the end of the letter), which could partly be due to our T cell culture conditions (see response to point 7). Furthermore, several reports in the literature show that a longer latency and tumor growth would be required to see any therapeutic effects, whereby the resulting large tumors would be in conflict with the ethical standards in Austria. Thus, the restrictions due to COVID-19, together with the lack of any efficacy even of the high-affinity control CAR excludes the generation of any meaningful data before the fall of this year.

Importantly, however, we did conduct another key *in vivo* experiment with the AND-gate AvidCAR in response to a comment from the other reviewer. In this new experiment, we mixed four different Nalm-6 populations (hEGFRt/hHER2t double-positive Nalm-6, both types of single-positive controls and double-negative versions) 1:1:1:1 and administered this mix of Nalm-6 cells intravenously to NSG mice. Thus, the specificity of the AND-gate AvidCAR for double-positive target cells vs. single positive or double negative controls could be analyzed within the same animal. In addition, we compared our AND-gate AvidCAR with a reference CAR containing the high-affinity version of the α EGFR binder (positive control) and T cells expressing no CAR (negative control). On day 13, i.e., 10 days after T cell inoculation, we determined the ratio and absolute cell counts of the four Nalm-6 populations in blood, bone marrow, spleen and liver.

Figs. 6c, 6d and Suppl. Fig. 10 show very convincingly that AvidCAR-T cells specifically killed the double-positive Nalm6 cells while ignoring the single-positive Nalm-6 cells, even though they were present in the same tissue of the same animal. Please note that this high specificity was obtained despite even higher antigen expression levels in the single positive controls (Suppl. Fig. 8d). In addition to the high specificity, the efficacy of T cells expressing the AND-gate AvidCAR was comparable to T cells expressing the high-affinity α EGFR-CAR (equally efficient in the liver; only slightly less efficient in bone marrow).

Together, this disseminated tumor model demonstrates that AND-gate AvidCARs are highly specific for double-positive cells and they do not require “safe distances”. It further shows that the efficacy of the AND-gate AvidCAR is comparable to the high-affinity control CAR. Although we agree that testing our AvidCARs in solid tumor models would have further improved our manuscript, we believe that it could not have answered the question of specificity and efficacy in a better way than the experiment that we are now showing in Figs. 6c, 6d and Suppl. Figs. 9-11.

Figure A. *In vivo* efficacy of CAR-T cells in NSG mice with subcutaneously growing tumor cells.

NSG mice (n=5) were implanted with hEGFR/hHER2-double positive K562 tumor cells (stably transduced cell clone, selected by limiting dilution assay for high expression of hEGFRt and hHER2t; 5×10^6 cells each on the left and right flank, s.c.). On day 7 the mice were treated with T cells (1×10^7 /mouse, i.v.) as indicated (AND-gate AvidCAR = α hEGFR/ α hHER2-AvidCAR). Mice treated with the α hEGFR-high-affinity CAR-T cells served as positive control. Tumor growth was measured twice weekly by digital caliper twice weekly and tumor volume was calculated. Shown are mean values \pm SEM (n=10 tumors in 5 mice).

Reviewers' Comments:

Reviewer #1:

Remarks to the Author:

Thank you - the authors have addressed my comments to my satisfaction. Very nice work.

Reviewer #2:

Remarks to the Author:

All the issues and concerns are addressed properly. I have no any further questions.

NCOMMS-19-28148A - Response to the remaining reviewers' comments

REVIEWERS' COMMENTS:

Reviewer #1 (Remarks to the Author):

Thank you - the authors have addressed my comments to my satisfaction. Very nice work.

We appreciate the reviewer's favorable comment on our revised manuscript.

Reviewer #2 (Remarks to the Author):

All the issues and concerns are addressed properly. I have no any further questions.

We appreciate the reviewer's favorable comment on our revised manuscript.